



# The impact of ship emissions on air quality and human health in the Gothenburg area – Part II: Scenarios for 2040

Martin O.P. Ramacher[1], Lin Tang[2,3], Jana Moldanová[2], Volker Matthias[1], Matthias Karl[1], Erik Fridell[2], Lasse Johansson[4]

[1]Helmholtz-Zentrum Geesthacht, 21502, Geesthacht, Germany
[2]IVL, Swedish Environmental Research Institute, P.O. Box 530 21, 40014 Gothenburg, Sweden
[3]WSP Environment Sweden, Box 13033, 402 51 Gothenburg, Sweden
[4]Finnish Meteorological Institute, P.O. Box 503, 00101 Helsinki, Finland

*Correspondence to*: Martin Ramacher (martin.ramacher@hzg.de)

**Abstract.** Shipping is an important source of air pollutants, from the global to the local scale. Ships are emitting substantial amounts of sulphur dioxides, nitrogen dioxides and particulate matter in the vicinity of coasts, threatening the health of the coastal population, especially in harbour cities. Reductions of emissions due to shipping have been targeted by several regulations. Nevertheless, effects of these regulations come into force with temporal delays, global ship traffic is expected to

grow in the future, and other land-based anthropogenic emissions might decrease. Thus, it is necessary to investigate combined impacts to identify the impact of shipping activities on air quality, population exposure and health-effects in the future.

We investigated the future effect of shipping emissions on air quality and related health effects considering different scenarios of the development of shipping under current regional trends of economic growth and already decided regulations in the Gothenburg urban area in 2040. Additionally, we investigated the impact of a large-scale implementation of shore electricity

in the port of Gothenburg. For this purpose, we established a one-way nested chemistry transport modelling (CTM) system from the global to the urban scale, to calculate pollutant concentrations, population weighted concentrations and health-effects related to $NO_2$, $PM_{2.5}$ and $O_3$.

The simulated concentrations of $NO_2$ and $PM_{2.5}$ in future scenarios for the year 2040 are in general very low with up to 4 ppb for $NO_2$ and up to 3.5µg/m³ $PM_{2.5}$ in the urban areas which are not close to the port area. From 2012 the simulated overall

exposure to $PM_{2.5}$ decreased by approximately 30 % in simulated future scenarios, for $NO_2$ the decrease was over 60 %. The simulated concentrations of $O_3$ increased from year 2012 to 2040 by about 20 %. In general, the contributions of local shipping emissions in 2040 focus on the harbour area but to some extent also influence the rest of the city domain. The simulated impact of wide use of shore-site electricity for shipping in 2040 shows reductions for $NO_2$ in the port with up to 30 %, while increasing $O_3$ of up to 3 %. Implementation of on-shore electricity for ships at berth leads to additional local reduction potentials of up to

3 % for $PM_{2.5}$ and 12 % for $SO_2$ in the port area. All future scenarios show substantial decreases in population weighted exposure and health-effect impacts.



## 1 Introduction

Shipping is an important source of air pollutants, from the global to the local scale. Nearly 70 % of ship emissions occur within 400 km of coastlines (Corbett et al., 1999), causing air quality problems through emissions of sulphur dioxide ($SO_2$), nitrogen dioxides ($NO_x$) and particulate matter (PM). An increase in shipping activity in the North Sea and the Baltic Sea has resulted

in higher emissions of air pollutants, and subsequently concentrations of pollutants in air, in particular of $NO_x$, especially in and around several major ports (Kalli et al., 2013). High contribution of shipping to emissions of sulphur and consequently to acid deposition and air pollution with particulate matter, mainly originating from oxidation of the $SO_2$ emissions, has been targeted by the International Maritime Organization (IMO) by setting limits on the maximum sulphur content of marine fuels (IMO, 2008). While on the global level a fuel sulphur content limit of 0.5 %, from previously 3.5 %,  has come into force on

January 1st 2020, the Baltic Sea and the North Sea have been declared sulphur emission control areas (SECA) in 2006 with gradually decreasing the fuel sulphur limit to 1.5, 1 and 0.1 % in 2006, 2010 and 2015, respectively (IMO, 2008). Additionally, a fuel sulphur limit of 0.1 % applies for ships at berth in all European harbours since 2010 (EU, 2005) and of 1.5 % for all passenger ships in regular line traffic in European waters outside the SECA (EU, 2012). To face the rising $NO_x$ emissions, the IMO has designated the North Sea and Baltic Sea as $NO_x$ Emission Control Areas (NECA) starting from January 1, 2021

onwards (IMO, 2017). The NECA regulation applies to all vessels built after 2021 and requires approx. 80 % $NO_x$ emission reductions (IMO, 2014). Due to the long lifetime of ships, it will take at least 30 years until the entire ship fleet will be renewed, which means that $NO_x$ emissions will only decrease gradually. In combination with the increasing ship traffic which grows roughly by 2 % per year and the future foreseeable significant decrease of emissions from other anthropogenic sectors (e.g. traffic, heating), the relative importance of $NO_x$ emissions from shipping for urban air quality will thus likely remain high.

To control emissions of greenhouse gases IMO has adopted a package of technical measures including the Energy Efficiency Design Index (EEDI). The EEDI regulation entered into force in 2013 and included requirements on minimum mandatory energy efficiency performance levels, increasing over time through different phases (IMO, 2011). In 2018 IMO adopted a resolution on the 'Initial IMO Strategy on reduction of GHG emissions from ships' stating the objective to reduce the total annual GHG emissions from international shipping by at least 50 % by 2050 compared to 2008 (IMO, 2018). Reaching this

objective implies both, efficiency gains and an increased use of renewable fuels. There is still a large potential for efficiency gains through better ship and engine design and through operational measures, mainly lower speeds. State of the art ships can be almost 50 % more efficient than ships that are 10-20 years old. Biofuels, wind power and electrification could play an important part in closing the gap between the potential of an improved engine design together with operational measures and the 50 % target for the entire sector, which, on the other hand, is expected to continue to grow in terms of transported volume

in the upcoming decades.

An important effect of the emission reductions of $SO_x$ and $NO_x$ is the resulting reduction in atmospheric concentrations of particulate matter (PM), especially secondary particulate sulphate and nitrate. Sofiev et al. (2018) have shown that the global limit on sulphur content in ship fuels decrease concentrations of particulate sulphate by 2-4 µg/m³ in the vicinity of busy ship



lanes on global scale, leading to significant reductions in $PM_{2.5}$ (particles with a diameter of less than 2.5 µm). The burden of $PM_{2.5}$ over the Baltic Sea region is predicted to decrease by 35 %-37 % between 2012 and 2040 as a result of regulation of $SO_x$ and $NO_x$ emissions and due to energy savings in shipping (Karl et al., 2019a). Importantly, the atmospheric transformation of $NO_x$ emitted from shipping is also relevant for ozone ($O_3$) formation (Eyring et al., 2010). The introduction of a NECA is

thus critical for reducing concentrations of $NO_2$ and $O_3$ at the same time.

In this study we investigate impacts of shipping on urban air quality and the associated health of the population in several future scenarios. We combine the development of emissions due to the implementation of the IMO rules on air pollutants and energy efficiency with changes in traffic volumes, fleet composition and fuel types used. In addition, the impact of a wide use of shore-side electricity by ships at berth is investigated. Only few studies considering impacts of shipping in future scenarios

specifically relevant for this region can be found in the literature (Cofala et al., 2018; Karl et al., 2019b; Karl et al., 2019a; Jonson et al., 2019; Jonson et al., 2015). However, the abatement measures considered as well as the methods used differ from our approach. The first part of our study (Tang et al., 2020) gives a brief overview of previous studies about impacts of shipping emissions on air quality and health at the Swedish west coast. It provides discussion on how the legislation changed between the base year used in our study (2012) and the situation today. Also, different methods of health impact assessment used in

these studies are briefly reviewed. In Tang et al. (2020) we discuss that shipping in Gothenburg in 2012 was a significant source of air pollution, contributing with 35 % and 12.5 % to the annual exposure to $NO_2$ and $PM_{2.5}$, respectively, and that the regional shipping outside the city was responsible for 20 % and 10 % of the $NO_2$ and $PM_{2.5}$ exposure, contributing more than the local shipping in and around the harbours. According to the study of Karl et al. (2019a), the introduction of the SECA with a fuel Sulphur limit of 0.1 % decreased the exposure to $PM_{2.5}$ at the Swedish West coast by approximately 35 %. This can be

seen as the regional part of the shipping contribution, because the maximum fuel sulphur content for ships at berth has been limited to 0.1 % in 2010, already. Sofiev et al. (2018) assessed the impact of the currently introduced global 0.5 % fuel sulphur content (FSC) limit on global scale in terms of health benefits and found that the introduction of the global 0.5 % FSC cap in 2020 leads to an avoidance of ~2'000 (5 % of cases due to shipping without the 0.5 % cap) premature deaths annually in Europe and ~ 137'000 (38 % of cases) globally. The impact at the West coast of Sweden was, however, found to be very small,

because North and Baltic Seas are SECAs with a maximum FSC of 0.1 % since 2015.

Cofala et al. (2018) assessed impacts of the implementation of emission control areas for $SO_x$ and $NO_x$ in all European seas in several alternative scenarios studying the years 2030, 2040 and 2050 and provided also cost-benefit analyses for these different alternatives. Different options of emission control areas in Southern Europe had very limited impact in Northern Europe, the study, however, also considered two different base scenarios, one of them including climate policy options for shipping.

Comparison of the data supplement Cofala et al. (2018) shows that the $PM_{2.5}$-related mortalities caused by shipping decreased in the 'climate measures' scenario compared to the no-climate-measures scenario by 0.8 % and 2 % (4 and 13 cases) in 2030 and 2050, respectively, in Sweden and by 1 % and 3.7 % (3 000 and 12 000 cases) in entire Europe. Cofala et al. (2018) also show the overall impact of shipping on the urban scale for the model grid cells including Mediterranean harbours. In the scenario without climate measures in 2030 the shipping contributions to annual mean $PM_{2.5}$ concentrations vary from ~0.2 to





µg/m³, while an introduction of additional SECA and NECA rules as in the North and Baltic Seas have the potential to avoid approximately 50 % of PM₂.₅. In 2050 the shipping contributions to PM₂.₅ were higher with concentrations up to 3 µg/m³. The reduction potential of SECA plus NECA introduction is about 65 %. This is more than in 2030 because the NECA effects come forward in a slow pace.

The health benefit of cleaner ship fuels and other emission reduction techniques in densely populated harbour cities is estimated to be much greater than on open sea. In order to quantify the future impact of shipping, scenarios for transported cargo volumes, composition of the fleet, as well as energy efficiency improvements need to be developed and put into perspective with probable emission reductions at land.

    The goal of the present study is to investigate the future effect of shipping emissions on air quality and related health effects
considering the development of shipping under current regional trends of economic growth and already decided regulations in the Gothenburg urban area in 2040. Additionally, we investigate the impact of a large-scale implementation of shore electricity in the port of Gothenburg. For this purpose, we established a one-way nested chemistry transport modelling (CTM) system from the global to the urban scale. This paper is the second part of a study about the current and future air quality situation in the Gothenburg urban area. Part 1 by Tang et al. (2020) is published in the same spacial issue.

## 2 Chemistry Transport & Health-Effect Modelling

### 2.1 Global- to urban-scale CTM system setup

    For the urban-scale, the prognostic meteorology-dispersion model TAPM (The Air Pollution Model, Hurley et al., 2005) was used as part of a one-way nested chemistry transport modelling (CTM) system from the global to the urban scale. TAPM has been successfully applied to investigate urban air quality and scenarios in coastal urban areas all over the world (e.g. Matthaios
et al., 2018, Ramacher et al., 2020, Gallego et al., 2016, Fridell et al., 2014).TAPM consists of a meteorological component and an air quality component. The meteorological component of TAPM is an incompressible, non-hydrostatic, primitive equation model with a terrain-following vertical sigma coordinate system for 3-D simulations. In the meteorological component, it is possible to assimilate wind observations to add a nudging term to the horizontal momentum equations. The air pollution component uses data from the meteorological component and consists of three modules: First, the Eulerian Grid
Module solves prognostic equations for mean and variance of concentrations, second, the Lagrangian Particle Module can be used to represent near-source dispersion more accurately, and third, the Plume Rise Module is used to account for plume momentum and buoyancy effects for point sources. The model also includes gas-phase reactions based on a Generic Reaction Set (Azzi et al., 1984) to represent the basic photochemical cycle of NO₂, NO and O₃, gas- and aqueous-phase chemical reactions for sulphur dioxide and particles, and a dust mode for total suspended particles (PM₂.₅, PM₁₀, PM₂₀ and PM₃₀). Wet
and dry deposition effects are also included.



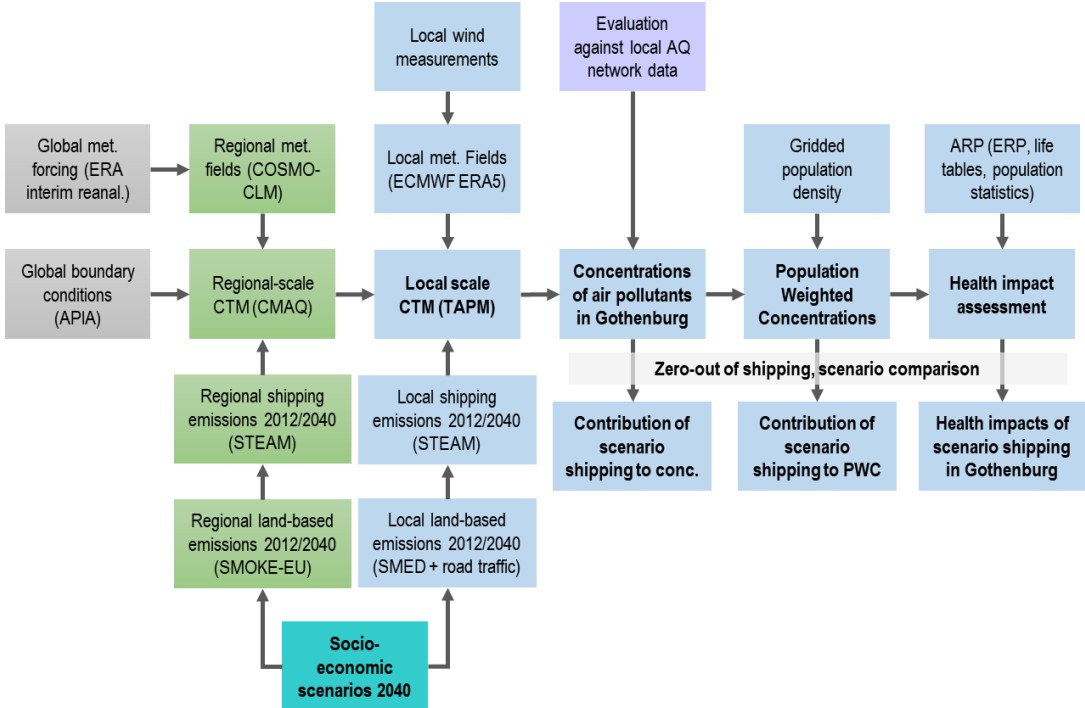

**Figure 1: Study design to calculate future concentrations, population exposure and health effects.**

### 2.2.1 Boundary conditions

For the Gothenburg urban area, we coupled TAPM offline to regional CTM simulations with the CMAQ model v5.0.1 (Byun

and Schere, 2006) as performed by Karl et al. (2019a) for the Baltic Sea region in 2012. Karl et al. (2019a) used global hemispheric pollutant concentrations from APTA global reanalysis (Sofiev et al., 2018) to consider global chemical boundaries and accounted for meteorological conditions with meteorological fields calculated for the COSMO-CLM mesoscale meteorological model version 5.0 (Rockel et al., 2008) for the year 2012 using the ERA-Interim re-analysis as forcing data (Geyer, 2014). Furthermore, they accounted for regional land-based emissions in 2012, represented by hourly gridded

emissions of $NO_X$, sulphur oxides, carbon monoxide (CO), $NH_3$, $PM_{2.5}$, coarse PM and non-methane volatile organic compounds (NMVOC) with the Sparse Matrix Operator Kernel Emissions for Europe model (SMOKE-EU, Bieser et al., 2011) Regional Shipping emissions for the Baltic Sea and North Sea with high spatial and temporal resolution were obtained from the Ship Traffic Emission Assessment Model STEAM (Jalkanen et al., 2009; Jalkanen et al., 2012; Johansson et al., 2013). Fridell et al. (2015) also accounted for future emission conditions with future scenarios for land-based and shipping emissions

in the Baltic Sea in 2040 which are consistent with the scenarios used in this study. Details of the regional air quality simulation setup including shipping emissions, the results and their evaluation in the 2040 Baltic Sea region are described in Karl et al. (2019a).





These simulations are used to interpolate chemical boundary conditions for TAPM. Concentrations simulated with CMAQ for the vertical model layer 7 with a mid-layer height of approximately 385 m above ground are used for this purpose. Since TAPM allows only one single boundary concentration value for the entire urban domain, these values are calculated every hour using horizontal wind components on each of the four lateral boundaries to give more weight to the concentrations upwind

the urban domain (Fridell et al., 2014). CMAQ simulations with and without ship emissions for 2012 and 2040 in the Baltic Sea and the North Sea were used as boundary conditions in the respective TAPM simulation runs with and without ship emissions for 2012 and 2040. This procedure allows for an analysis of regional influences on the Gothenburg area.

### 2.2.2 Meteorological fields

The spatial resolution of the urban domain for the TAPM air pollution component is 250m x 250m. With an extent of 25 km

x 25 km this domain is covering the city of Gothenburg and the harbour area along the shores of the Göta river running through the city. The urban domain for the TAPM air pollution component is nested in 30 km x 30 km (500 m horizontal resolution) hourly meteorological fields taken from the innermost domain of nested simulations with the meteorological component of TAPM. We chose a smaller domain for the TAPM air pollution component, because of a higher efficiency in computing time while having all important city features covered.

TAPM includes a nested approach for meteorology, which allows to zoom-in to a local region of interest, while the outer boundaries of the grid are driven by synoptic-scale analyses. We applied the meteorological component with four nested domains from 480 x 480 km² extent at the outer domain (D1) to 30 x 30 km² extent at the inner domain (D4). The outer domain (D1) was forced by ECMWF ERA5 synoptic meteorological reanalyses ensemble means with 30 vertical layers, $0.3° \times 0.3°$ horizontal and three-hourly temporal resolution. Additionally, hourly local wind fields of four measurement stations (Femman,

Gothenburg, Landvetter, Vinga) operated by the Swedish Meteorological and Hydrological Institute (SMHI) have been assimilated in the meteorological component to force the meteorological fields to be closer to the measurements. Since this study focuses on the impact of changes in shipping emissions in 2040, and not on meteorological effects, the 2040 simulations also use 2012 meteorological fields. Details on meteorological and chemical component configurations in TAPM as well as air quality results and their evaluation for 2012 can be found in the accompanying paper by Tang et al. (2020).

### 2.2.3 Current and future land-based emission inventories

A bottom-up emission inventory for the Gothenburg urban area has been created to account for road traffic, industrial processes and other sources of land-based emissions in 2012. The road traffic emissions are calculated with traffic activity data from the database of the Environmental Administration, City of Gothenburg (Miljöförvatningen) and a set of emission factors for the Swedish road vehicle fleet in 2035 (latest year available) by HBEFA v. 3.2 (Hand Book of Emission Factors for Road

Transport, Keller et al., 2017). The traffic sources are treated as line emission sources in TAPM. For the ten biggest industrial sources, emission fluxes assigned with coordinates and emission heights were obtained from the Swedish Environmental Emission Data (SMED) for 2012 and modelled as area sources in TAPM. Remaining sources, which are non-road activities,





waste and sewage, domestic heating, energy production, combustion in industry for energy purposes, non-road working machinery, domestic aviation, solvents from product use and agriculture, are gathered from the SMED gridded inventory. They are geographically distributed on a 1km x 1km grid and modelled as gridded area sources. The land-based emission inventory created in this way takes into account all relevant emission sources for $SO_2$, $NO_x$, $PM_{10}$, $PM_{2.5}$ and VOCs in 2012.

For land-based emissions in 2040, the 2012 emission inventory was scaled to 2040 conditions using source specific approaches. The road traffic emission inventory in 2040 uses detailed activity data for 2012 scaled with a traffic volume development scenario in Sweden specific for light and heavy-duty vehicles and busses (Transport administration, 2016, 2018). Combined with emission factors, which were calculated for the expected Swedish car fleet in 2035 using HBEFA v.3.2 database (year 2040 was not available), a road traffic emission inventory for 2040 was calculated. The annual road traffic emissions in 2040

are with 265t NOx / year, 193t VOC / year and 141t PM 10 / year about 89 %, 62 % and 12 % lower than in 2012 in the Gothenburg area (Fig. 2).

The area emissions, covering all emission sectors except road traffic and shipping, are scaled from 2012 to 2040 with factors which describe the change in Swedish emissions between 2010 and 2040. These factors were calculated with emissions obtained from the Greenhouse Gas - Air Pollution Interactions and Synergies (GAINS) model using the emission scenario

ECLIPSE_V5a_CLE_base (Kiesewetter et al., 2014) and can be found in Supplement 1. The scenario emissions for the time-period 2000 - 2040 were provided as national emissions for European countries, including Sweden, specified for GAINS emission sector categories by Zbigniew Klimont, IIASA (personal communication). These categories were translated into the emission categories of Swedish Environmental Emission Data (SMED) and the 2040/2010 factors were applied to the 2012 area emission sectors to derive a land-based emissions scenario for 2040 (CLE2040). The year 2012 is not available for

ECLIPSE_V5a_CLE_base but the change between 2010 and 2012 is small on the 30-years horizon. Based on these factors, the annual industrial emissions in 2040 are with 468t NOx / year, 9958t VOC / year, 85t PM 10 / year and 189t SO2 / year about 19 % lower, 45 % higher, 7 % lower and 2 % lower than in 2012 in the Gothenburg area (Fig. 2). The reason for an increase in VOC emissions in the future is a scaling factor of 1.45 for the sector 'Combustion in industry for energy purposes', which six out of ten industrial sources in the Gothenburg area belong to. As part of the CTM chain, the treatment of area

emissions for the urban area of Gothenburg is consistent with the method used in the regional-scale CMAQ simulations. Thus, the boundary conditions in the local TAPM runs were taken from corresponding regional-scale simulations of the CMAQ model with consistently derived emissions for 2012 and 2040 (Karl et al., 2019a).

**2.3 Exposure & Health impact assessment**

The impacts of exposure to air pollutants on the health of people living in the Gothenburg region were assessed with the

ALPHA-RiskPoll model (ARP, Holland et al., 2013) which calculates a wide range of air-pollutant specific health effects in the assessed year. The RAINS methodology which calculates years of life lost over the expected lifetime of a population (Amman et al., 2004) has been used as well to enable a comparison with other studies. Both methods are based on national population statistics for European countries and on a forecast of the age distribution of the population, as well as mortality and





morbidity data for 2040. In addition, effect-specific dose response relationships are taken into account. In case of the RAINS methodology only all-cause mortality from $PM_{2.5}$ exposure has been considered. In the ARP analysis, impacts of exposure to $PM_{2.5}$, ozone and $NO_2$ have been considered (Heroux et al., 2013). Only the most serious impacts, i.e. losses of lives, are presented, taking into account impacts of chronical exposure to $PM_{2.5}$, short-term exposure to ozone and short-term exposure

to $NO_2$, i.e. the impacts marked A* in the HRAPIE (Health risks of air pollution in Europe) study (Heroux et al., 2013). For ozone, the indicator SOMO35 is used, standing for the annual sum of the daily maximum of the 8-hours mean ozone concentrations above a threshold of 35 ppb. The health impacts of some pollutants are correlated and that is why the premature deaths attributed to each pollutant cannot simply be added up. The concentration-response functions (CRF) for all-cause mortality used in ARP are those from WHO (Heroux et al., 2013), 6.2 % (95 % confidence interval 4.0 % - 8.3 %) relative risk

increase per 10 µg/m$^3$ increased exposure for the long-term $PM_{2.5}$ exposure, 0.29 % (95 % confidence interval 0.14 % - 0.43 %) relative risk increase per 10 µg/m$^3$ increased exposure for the short-term ozone exposure and 0.27 % (95 % confidence interval 0.16 % – 0.38 %) relative risk increase per 10 µg/m$^3$ increased exposure for the short-term $NO_2$ exposure. The RAINS methodology uses 5.8 % relative risk increase per 10 µg/m$^3$ increased exposure to $PM_{2.5}$. More details on the methodology can be found in Part 1 of these papers (Tang et al., 2020).

The exposure calculation was based on the concentration fields of $PM_{2.5}$, $O_3$ and $NO_2$ calculated for the examined future scenarios by the modelling system described above. Annual means and SOMO35 were calculated from hourly ozone concentration fields. Population data at 1km × 1km resolution were obtained from Statistics Sweden (SCB) for 2015 with a population of 572'779 in the city of Gothenburg and used for calculating the population weighted average concentrations (PWC) for the model domain in 2012 (Tang et al., 2020). For the 2040 scenarios the PWC were calculated using the same

population data for 2015 since a geographically resolved prognosis for 2040 was not available. In ARP the PWCs are applied on the population statistics for Sweden for the year 2040 and scaled to the population of Gothenburg with help of the year-2012 'Gothenburg population'/'total Swedish population' ratio. This approach neglects any potential trend of increase of urbanization in the country which would lead to higher impacts than calculated with our approach.

With the introduced study design, it is possible to estimate the impact of shipping-related air pollution on the health of citizens

in the Gothenburg area regarding current and future emission scenarios and to identify the effectiveness of several air pollution abatement measures. For this purpose, it is necessary to create a set of scenarios with emphasis on shipping activities in the future, translate them into emission inventories and simulate the health effects with the introduced CTM-exposure-health effect modelling system.

## 3 Current and future shipping emissions scenarios

### 3.1 Ship emission inventories for the Gothenburg area

A shipping emission inventory for the area of Gothenburg with high temporal and spatial resolution was calculated with STEAM for the year 2012, representing the present situation (Tang et al., 2020) and giving a baseline to be compared with





future scenarios. In STEAM, position data of individual ships taken from reports from the Automatic Identification System (AIS) is used to model fuel consumption and emissions as a function of vessel activity, engine and fuel type. The calculation of ship emission inventories for the Gothenburg area follows the approach that has been applied for the North and Baltic Sea region which is described in Karl et al. (2019a). The emission inventory in this work is therefore consistent with the one in

Karl et al (2019a). Nevertheless, the regional shipping emission inventory contains hourly updated emission data on a 2 km × 2 km grid, while the local emission inventory comes with a resolution of 250 m × 250 m for the local research domain. The ship emissions in the Gothenburg area include combustion emissions from all ship engines (boilers, auxiliary and main engines) for the compounds $NO_x$, $SO_x$, CO, $CO_2$, NMHC and PM. Tang et al. (2020) used STEAM shipping emission inventory in the Gothenburg area and applied it in the presented global-to-local CTM system to identify the impact of shipping on urban air

quality in the year 2012. In 2012, the local ship emissions in Gothenburg hold with 308 t $SO_2$ / year, 2089 t $NO_x$ / year, 91 t $PM_{10}$ / year and 23 t VOC / year for about 60 % of $SO_2$, 40 % of $NO_x$, 25 % of $PM_{10}$ and 1 % of VOC respectively, to the total emission situation (Fig. 2). Thus, shipping emissions are a major contributor to the urban air quality in Gothenburg in 2012.

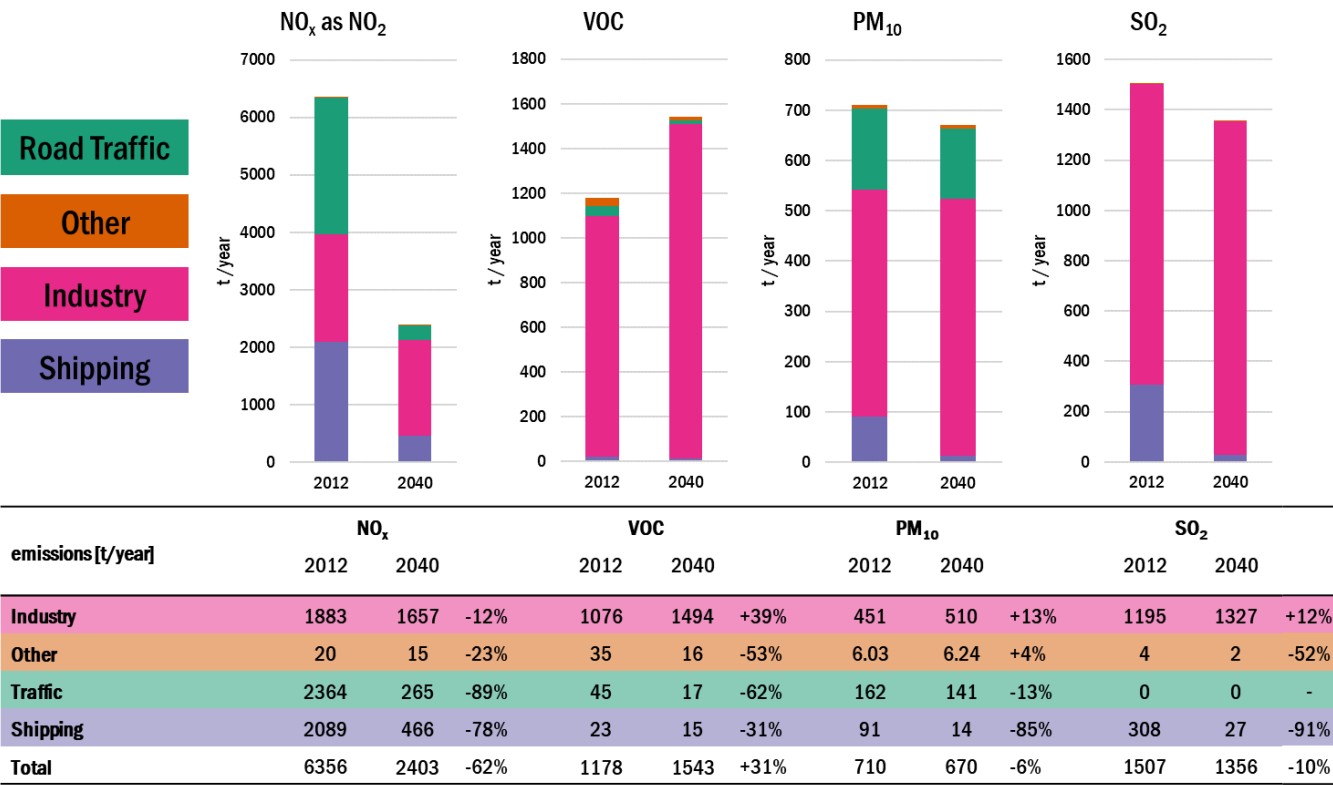

| emissions [t/year] | $NO_x$ | | | VOC | | | $PM_{10}$ | | | $SO_2$ | | |
|---|---|---|---|---|---|---|---|---|---|---|---|---|
| | 2012 | 2040 | | 2012 | 2040 | | 2012 | 2040 | | 2012 | 2040 | |
| Industry | 1883 | 1657 | -12% | 1076 | 1494 | +39% | 451 | 510 | +13% | 1195 | 1327 | +12% |
| Other | 20 | 15 | -23% | 35 | 16 | -53% | 6.03 | 6.24 | +4% | 4 | 2 | -52% |
| Traffic | 2364 | 265 | -89% | 45 | 17 | -62% | 162 | 141 | -13% | 0 | 0 | - |
| Shipping | 2089 | 466 | -78% | 23 | 15 | -31% | 91 | 14 | -85% | 308 | 27 | -91% |
| Total | 6356 | 2403 | -62% | 1178 | 1543 | +31% | 710 | 670 | -6% | 1507 | 1356 | -10% |

**Figure 2: Present day and future 2040BAU scenario emission inventories for the local CTM simulation in Gothenburg.**



### 3.2 Future scenarios for shipping emissions

The scenarios used in this work describe future developments of policy and technology regarding energy efficiency and exhaust gas emissions from ships in the North and Baltic Sea region as well as in the port of Gothenburg all taking into account:

- the development of ship traffic and transport volumes,
- fleet development of different ship types,
- changes in fuel mixture,
- the use of abatement measures and other technologies that influence emissions from shipping,
- regulations influencing emissions and fuel consumption,
- possible local port actions in Gothenburg, e.g. use of shore-side electricity for ships at berth.

The scenarios were created in the BONUS SHEBA project and are based on literature reviews and expert and stakeholder consultations to assess shipping in the future within different developments (Fridell et al., 2015; Karl et al., 2019a). The overall goal was to investigate changes in impacts of shipping on the marine and terrestrial environment as well as on human health in the Baltic Sea region. The scenario results have been used to assess urban-scale impacts on air quality and human health in the Gothenburg area and several other Baltic Sea harbour cities (Ramacher et al., 2019a). In this work shipping in the urban

area of Gothenburg in the future is modelled in four scenarios for 2040:

- BAU2040 – Business as usual 2040, this scenario is the future reference scenario including all currently adopted regulations including climate measures with high energy improvements in energy efficiency (Kalli et al., 2013) (still not achieving IMO 2018 Initial Strategy to reduce $CO_2$ emissions by 50 % relative to 2008 by year 2050).
- BAU2040LP – BAU2040 with additional implementation of shore-side electricity.
- EEDI2040 – As BAU2040 but fuel efficiency just follows energy efficiency design index regulation of the IMO.
- EEDI2040LP – EEDI2040 with additional implementation of shore-side electricity.

### 3.2.1 Future reference scenario BAU2040

The BAU2040 scenario is based on current trends in shipping and takes into account already decided policy measures (Table 1). This represents a conservative development of shipping in line with the Shared Socioeconomic Pathways (SSP) II "Middle

of the Road" scenario (Zandersen et al., 2019), which is developed for the climate community and adapted for shipping in the Baltic Sea. The trends in shipping were analysed from AIS data from recent years and combined with an analysis of the different shipping sectors to obtain the development regarding transport work, ship size, ship speed and number of ships for different ship types as done for the regional-scale by Karl et al. (2019a). In combination with assumptions on ship age distribution and upcoming regulations (Fridell et al., 2015), this allows for the calculation of emissions to air. The following

regulations affecting emissions to air were applied in BAU2040 (Table 1):

1. Sulphur regulation: The Baltic and North Seas are Sulphur Emission Control Areas (SECA) where the maximum allowed sulphur content in marine fuel was lowered from 1 % to 0.1 % in 2015. For sea areas outside SECAs the





maximum fuel sulphur content is 0.5 % from 2020. For ships berthing in EU ports the maximum allowed fuel sulphur content is 0.1 %, these regulations directly influence the emissions of $SO_X$ and have a strong impact on the PM emissions. These regulations are also applied in the EEDI2040 scenarios.

2. $NO_x$ regulation: $NO_x$ emissions from marine engines are regulated with Tier I for new ships since 2000 and Tier II
since 2011. Tier III is applied in $NO_x$ Emission Control Areas for new ships operating in the Baltic and North Seas from 2021. These regulations are also applied in the EEDI2040 scenarios.

3. Energy efficiency: The regulation by IMO on Energy Efficiency Design Index (EEDI) (IMO, 2018) requires new ships to become gradually more fuel-efficient. The improvements in energy efficiency, fuel use reductions and emissions are assumed proportional.

The BAU2040 scenario assumes a share of ships driven by liquefied natural gas (LNG) of about 10 % in the ship fleet in 2040. This is modelled as a fraction of new ships introduced each year that will use LNG since retrofitting of existing ships from fuel oil to LNG is assumed less likely due to high costs. Since LNG is used as a means to comply with the sulphur and $NO_x$ regulations, ship types that operate mainly within SECAs are modelled as being more likely to use LNG. The BAU2040 scenario also assumes that on average 20 % of the ships in the Baltic Sea use scrubbers. This measure, however, does not affect
emissions to air in our study since the scrubbers are required to reach SOX emissions equivalent to using MGO and that the PM emissions are similar as for MGO (Fridell and Salo, 2016). The energy efficiency for new ships in BAU2040 is assumed to improve further than what is required from the EEDI regulation, following recent trends and assumptions from Kalli et al. (2013), assuming annual efficiency increases of 1.3 % to 2.25 %, depending on ship type (corresponding efficiency increase values required by the IMO EEDI regulation is 0.65 % to 1.04 %) which significantly reduces shipping fuel consumption.

**Table 1. Major regulation changes for the different scenarios.**

| Scenario | FSC in global | FSC in SECA area | FSC in Gothenburg area | NOx regulation in NECA | NOx regulation in Gothenburg area | LNG | Scrubbers |
|---|---|---|---|---|---|---|---|
| 2012 reference | 3.5 % | 1.0 % | 0.1 % | Tier II standard | Tier II standard | / | / |
| 2040 scenarios | 0.5 % | 0.1 % | 0.1 % | Tier III standard | Tier III standard | 10 % | 20 % |

Based on these assumptions scaling factors 2040/2012 were calculated by applying fleet development, fuel mix, abatement technology implementation and improvements in energy efficiency trends on fleet composition in the Gothenburg area
calculated with STEAM for the year 2012. These have been applied on the 2012 gridded shipping emissions inventory to calculate the BAU2040 emission scenario. Compared to the present situation in 2012, the annual shipping emissions in BAU2040 are decreased to 466 t NOx / year (-78 %), 23 t VOC / year (-31 %), 91 t $PM_{10}$ / year (-85 %), 27 t $SO_2$ / year (-91 %) (Fig. 3). While the shipping emissions in 2012 have been a major contributor to the overall air pollution (Fig. 2), in 2040 the relevance of shipping emissions decreases in comparison to industry, road traffic and other sources to 19 % for NOx, 1 %
for VOC, and 2 % for $PM_{10}$ and $SO_2$.





### 3.2.2 Future scenario EEDI2040

In the EEDI2040 scenario, improvements of fuel efficiency follow strictly the requirements of the EEDI (Energy Efficiency Design Index) regulation of the International Maritime Organization. Annual efficiency increases of 0.65 % to 1.04 %, depending on ship type, are assumed in the EEDI2040 scenario while the corresponding values in the BAU2040 scenario are

1.3 % to 2.25 %. From the difference between BAU2040 and EEDI2040 the effect of the higher fuel efficiency increase than required by the EEDI regulation, can be deduced.

Based on these assumptions, scaling factors have been calculated in the same manner as for the BAU2040 scenario and applied to the 2012 shipping emissions inventory. Compared to the present situation in 2012, the annual shipping emissions in EEDI2040 are decreased to 666 t NOx / year (-68 %), 22 t VOC / year (-2 %), 19 t $PM_{10}$ / year (-79 %), 38 t $SO_2$ / year (-88

%) (Fig. 3). In comparison to 2012, the relevance of shipping emissions in the EEDI2040 scenario decreases in comparison to industry, road traffic and other sources to 26 % for NOx, 1 % for VOC, and 3% for $PM_{10}$ and $SO_2$.

### 3.2.3 Future shore-side electricity scenarios - BAU2040LP and EEDI2040LP

In addition to regional developments and regulations, which are reflected in the BAU2040 and EEDI2040 scenarios, large-scale implementation of shore-side electricity (or land power, LP) in the port of Gothenburg was studied in both scenarios.

Concerns about air quality in port cities as well as policies on greenhouse gas emissions have led to measures aiming at reducing the use of auxiliary engines by ships at berth and thereby reducing emissions of air pollutants and greenhouse gases as well as noise through use of shore-side electricity.

A gridded emission inventory with large-scale shore-side electricity use in 2040 was calculated with STEAM in the following way: All RoRo, RoPax, and cruise ships and 50 % of all other ships use shore-side electricity. Scaling factors have been

calculated in the same way as for the future scenarios BAU2040 and EEDI2040, except that emission from ships at berth were reduced as described. These factors were then applied in the gridded emission inventories, resulting in 165 t $NO_x$ / year, 5 t VOC / year, 5 t $PM_{10}$ / year and 17 t $SO_2$ / year emissions in Gothenburg in the BAU2040LP scenario (Fig. 3). The EEDI2040LP annual shipping emissions result in 234 t NOx / year, 7 t VOC / year, 7 t PM10 / year and 24 t SO2 / year. Compared to the BAU2040 and EEDI2040 scenario, the annual emissions in both, the BAU2040LP and EEDI2040LP scenario, are 65 % lower

for $NO_x$, 68 % lower for VOC, 62 % lower for $PM_{10}$ and 37 % lower for $SO_2$. While the shipping emissions in 2012 have been a major contributor to the overall air pollution (Fig. 2), in 2040 the relevance of shipping emissions decreases in comparison to industry, road traffic and other sources to 19 % for NOx, 1 % for VOC, and 2 % for $PM_{10}$ and $SO_2$ in the BAU2040LP scenario.





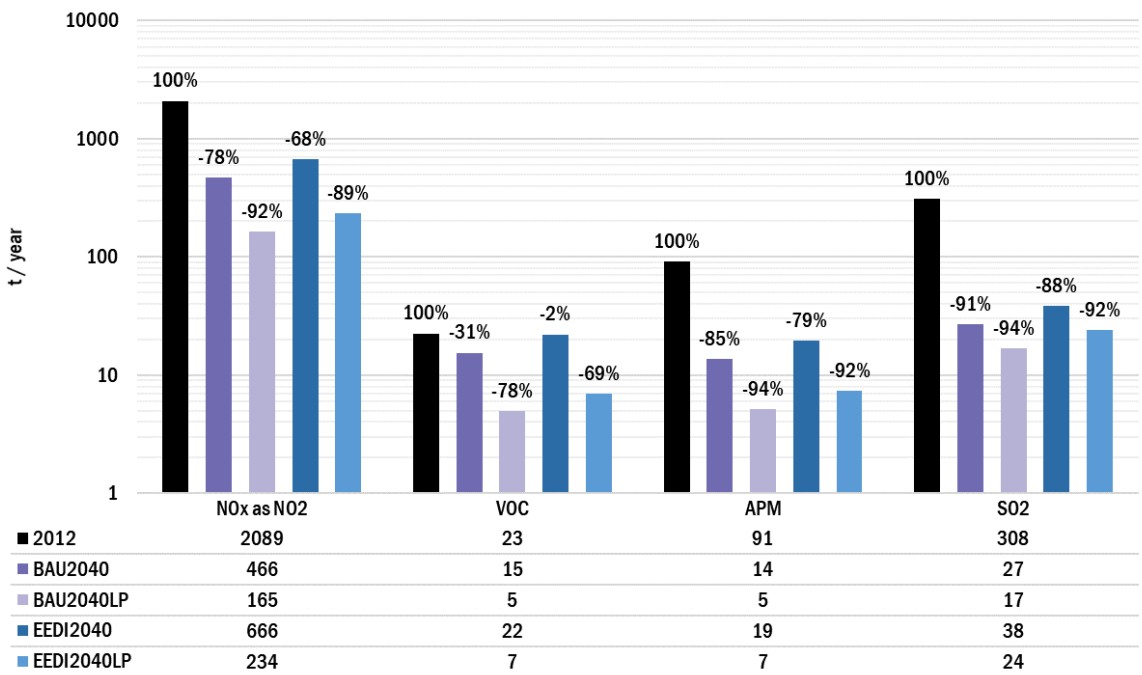

| | NOx as NO2 | VOC | APM | SO2 |
|---|---|---|---|---|
| ■ 2012 | 2089 | 23 | 91 | 308 |
| ■ BAU2040 | 466 | 15 | 14 | 27 |
| ■ BAU2040LP | 165 | 5 | 5 | 17 |
| ■ EEDI2040 | 666 | 22 | 19 | 38 |
| ■ EEDI2040LP | 234 | 7 | 7 | 24 |

**Figure 3: Annual Shipping emissions of NO$_x$, VOC, PM$_{10}$ and SO$_2$ for the local CTM simulation in Gothenburg for 2012, BAU2040 and BAU2040LP emissions scenarios.**

### 3.3 Scenario Setup

The introduced land-based (CLE2040) and shipping (BAU2040, BAU2040LP, EEDI2040 and EEDI2040LP) emission inventories for 2040 have been applied in the established global-to-local CTM system to identify:

(1) the impact on air quality in Gothenburg through a change in total emissions from 2012 to 2040,

(2) the impact of local shipping activities in 2040 in two different scenarios, and

(3) the additional impact of local port measures (shore-side electricity) in scenarios for 2040.

The meteorological conditions were held constant for all regional and local CTM runs as our focus is on the impact of changing emissions. The regional boundary conditions applied to the local-scale TAPM simulations for the BAU2040LP and EEDI2040LP were taken from regional CTM simulations with BAU2040 and EEDI2040 emissions including shipping emissions (Table 1). By using the local land-based emissions in line with the regional land-based emissions and varying the local shipping emissions, this scenario setup allows for the assessment of local shipping impacts in different local scenarios.

To derive the contribution of ships to the selected pollutant concentrations, two model runs for each scenario, one including and one excluding local shipping emissions in TAPM simulations, were performed. The difference is regarded as the contribution of ships to the individual pollutant. For the scenarios, the difference between two model runs with different shipping emissions is regarded as the change in the contribution of ships between the respective scenarios. In the discussion





of the results, the BAU2040 scenario will be discussed as the future reference scenario. Consequently, we will show results for the BAU2040 scenario in the main paper, while results for EEDI2040 are available in Supplement 2.

**Table 1: Meteorology, regional boundary conditions and emissions setup for the calculated scenarios.**

| Scenario | Meteorology | Reg. Boundary | Local Emissions | |
|---|---|---|---|---|
| | | | **Land based** | **Shipping** |
| 2012 Reference | 2012 | 2012 (incl. shipping) | 2012 | 2012 |
| BAU2040 | 2012 | BAU2040 (incl. shipping) | CLE2040 | BAU2040 |
| BAU2040LP | 2012 | BAU2040 (incl. shipping) | CLE2040 | BAU2040LP |
| EEDI2040 | 2012 | EEDI2040 (incl. shipping) | CLE2040 | EEDI2040 |
| EEDI2040LP | 2012 | EEDI2040 (incl. shipping) | CLE2040 | EEDI2040LP |

## 4 Future impact of shipping on concentrations of pollutants

BAU2040 serves as reference scenario all other scenarios are compared to. It was first compared to the present-day air quality situation in 2012, which is discussed in detail in the accompanying paper by Tang et al. (2020). Ship activities in Gothenburg 2012 contribute to peak values, in particularly in the north of the city port and the river Göta due to the dominant SW wind. The local shipping contribution to $NO_2$ concentrations in Gothenburg was about 14 % (around 0.5 ppb) to the annual mean averaged over the entire model domain. These contributions were higher in summer due to higher ship activities. Emissions of $NO_x$ from ships added up with land-based $NO_x$ emissions and enhanced the local ozone loss by NO titration. The negative effect of $NO_x$ emissions from local shipping on $O_3$ concentrations in summer was -2 % (around -0.5 ppb) on average. For $PM_{2.5}$, the local ship emissions contributed about 2 % (around 0.07 µg/m$^3$) to the annual mean, while the annual average $SO_2$ concentrations from local shipping were the major contributor to local $SO_2$ emissions with 0.2 - 0.5 ppb along major shipping lanes. In the following, maps illustrating changes in annually averaged concentrations of $NO_2$, $O_3$, $PM_{2.5}$ and $SO_2$ are shown for the total change in ambient air concentrations from 2012 to 2040BAU (Fig. 4), the impact of change in shipping emissions from 2012 to BAU2040 on shipping contribution to ambient concentrations of these air pollutants (Fig. 5) as well the impact of a large-scale introduction of shore-side electricity in the BAU2040LP and EEDI2040LP scenarios on the contribution of shipping compared to BAU2040 and EEDI2040, respectively (Fig. 6 for BAU2040 results, Supplement 3 for EEDI2040 results). Seasonal plots for summer and winter months can be found in Supplements 2 and 3.

### 4.1 Air quality changes in 2040 compared to present-day

The local concentration of $NO_2$, given as annual average over the model domain, decreased by 74 % (around 2.8 ppb) from about 3.7 ppb in 2012 to 0.9 ppb in the future reference scenario BAU2040. The highest changes in $NO_2$ are located in the centre of Gothenburg with an average $NO_2$ reduction of up to 80 % (and 8 ppb) next to major roads. Besides the high reductions due to road traffic, reduction of $NO_2$ concentration due to the reduction of emissions from industrial sources is visible in the

-1





western part of the Gothenburg domain with reductions of up to 7 ppb (~30 %). The smaller relative reduction in industrial areas, is due to the comparatively low change of industrial $NO_2$ emissions and their already high contribution to $NO_2$ concentration in the western part of the city in 2012 on one side, and the high reduction in road traffic emissions and a high density of highways and road traffic in the eastern part of the Gothenburg domain on the other side. The port area, which is

located westward of the centre, shows a comparably high reduction potential with up to 7 %. The nearby industrial sources might hide reductions in $NO_2$ from other sources due to their high absolute contributions and relatively low reduction from 2012 to BAU2040.

When it comes to changes in $O_3$ concentrations, there is an increase in the city centre by up to 15 % (about 4 ppb) from 2012 to BAU2040, especially near major roads. This contrary trend follows the principles of ground-level ozone formation, which

is produced in photochemical reaction cycles involving the precursors $NO_x$ and VOCs. The ozone-precursor relationship in urban environments is a consequence of the fundamental division into a NOx-sensitive and a VOC-sensitive regime (Sillman, 1999). VOC-sensitive regimes in dense urban areas with many emission sources, lead to higher $O_3$ with increasing VOC and lower $O_3$ with increasing $NO_x$ (Karl et al., 2019a). Therefore, the contribution of local ship emissions with ozone precursors $NO_x$ and VOCs can selectively be very significant, in terms of both, increasing the $O_3$ levels in urban areas and decreasing

them in the outskirts. Fig. 4 shows that both, the absolute and the relative change in impact of shipping activities between 2012 and BAU2040 becomes more visible for $O_3$ than for $NO_2$ in the western parts of the city, due to the higher ozone formation in the absence of $NO_x$ sources in the BAU2040 scenario.

For $PM_{2.5}$ and $SO_2$ the high impact of some industrial sources in the West of Gothenburg is even more visible. While in 2012 the average $PM_{2.5}$ concentrations peak at 53 µg/m³ in the vicinity of the largest point sources, in BAU2040 they peak at 48

µg/m³. The domain averaged $PM_{2.5}$ concentrations are much lower with 4 µg/m³ in 2012 and 2.7 µg/m³ in BAU2040, thus they are reduced by 33 % in average. Slightly higher reductions close to roads are caused by lower $PM_{2.5}$ road traffic emissions. Reductions in the Northeast of the urban area of Gothenburg are probably due to less secondary particle formation. This pattern holds also true for $SO_2$. There is an absolute reduction potential for $SO_2$ of up to 1 µg/m³ and a relative reduction potential of up to 75 % in the port area, following shipping routes. Nevertheless, the characteristic industrial point sources bear the highest

absolute $SO_2$ reductions and therefore partly diminish the relative $SO_2$ reduction potential in the port area of Gothenburg.

In total, the air quality situation with respect to $NO_2$, $PM_{2.5}$ and $SO_2$ is clearly improving in the urban area of Gothenburg in the BAU2040 scenario. However, the large industrial point sources such as three refineries (Preem Gothenburg, St1 Refinery AB, Nynäs Gothenburg) are identified as large contributors to spatially selective high concentrations of $NO_2$, $PM_{2.5}$ and $SO_2$ and still contain a high reduction potential compared to all other sources of air pollution in the urban area of Gothenburg.

When it comes to ozone there is an average increase of up to 1 ppb in summer, probably due to a lower background concentration and consequently less ozone titration by the lower $NO_x$ emission in 2040.





**Figure 4: The total modeled present day concentration for NO₂, O₃, PM₂.₅ and SO₂ (column 1), as well as the concentration in BAU2040 (column 2) and the difference between the present day and BAU2040 in absolute (column 3) and relative (column 4) values. © OpenStreetMap contributors 2019. Distributed under a Creative Commons BY-SA License.**



## 4.2 Influence of ship emissions in the future scenarios: BAU

The modelled contributions of local shipping to atmospheric concentrations and relative contributions to the overall air pollution in Gothenburg in the BAU2040 scenario show high reductions relative to the year 2012 for all pollutants under investigation except of $O_3$, which is slightly increasing (Fig. 5). Higher absolute and relative contributions of local shipping are detected in and around the port area, while there are some minor impacts in the Northern urban area of Gothenburg due to predominant winds from Southwest. This general pattern also holds true for the EEDI2040 scenario. The maximum value for the contribution to annual mean $NO_2$ concentrations in BAU2040 merely reaches 1 ppb in the port area and is about 80 % lower compared to a maximum of 4.1 ppb in 2012. In the EEDI2040 scenario the maximum ship contribution to $NO_2$ is slightly higher with 1.4 ppb. The relative contribution of shipping given as annual average in the entire model domain changed from 14 % to 6 % in BAU2040. In the EEDI2040 scenario, the relative contribution of local shipping to the annually averaged grid means reaches 18 %. The relative contributions of shipping in the port area of Gothenburg is up to 25 % in BAU2040 and up to 45 % in EEDI2040. In 2012, $NO_2$ concentrations due to shipping are involved as precursor in the photochemical reaction-cycle of $O_3$ formation and form a depletion pattern around the harbour area with up to -4 ppb $O_3$. Following the principles of $O_3$ formation in a high-$NO_x$-environment the pattern shows ozone formation from shipping emissions further from the harbour area. The same pattern is visible in future scenarios but with only a small depletion of -0.5 ppb $O_3$ at maximum in BAU2040 and -1.2 ppb $O_3$ in EEDI2040. While in 2012, $O_3$ concentrations increase by up to 8 % (~2 ppb) outside the port area, in the future scenarios shipping related $O_3$ concentrations are on average around 0, except for the area with industrial emission sources in the West. Here, high VOC emissions from the industrial sources react with $NO_x$ emissions from nearby shipping and form about 1 ppb $O_3$ at maximum, which can be accounted to shipping activities. Nevertheless, the overall contribution of shipping to increased $NO_2$ and $O_3$ concentrations is very low in both future scenarios.

The pollutants $PM_{2.5}$ and $SO_2$ show similar reduction patterns in the future scenarios. The huge reductions in $PM_{2.5}$ (-85 %) and $SO_2$ (-91 %) emissions are consequently leading to a reduced impact of shipping in BAU2040. The contribution of $PM_{2.5}$ from local ship emissions is relatively low in 2012 (maximum of 0.9 µg/m³ in the western port area), and even lower (maximum of 0.15 µg/m³ in the western port area) in the BAU2040 scenario. The $SO_2$ concentrations in the Gothenburg area are driven by industrial and shipping emissions, which account for more than 99 % of the total, both in 2012 and in BAU2040. Between 2012 and BAU2040 the $SO_2$ emissions from shipping decreased by 91 % and therefore the concentration of $SO_2$ decreased as well. While there has been a relative contribution of shipping to $SO_2$ concentrations in summer of about 70 % in the harbour and its surrounding areas in 2012 (concentration contribution maxima of up to 0.7 µg/m³), in the 2040BAU scenario the contributions are below 20 % with maximum concentration contribution of less than 0.2 µg/m³ to the summer mean. To summarise, the air pollution from shipping in the BAU2040 scenario reflects the large emission reductions compared to 2012, resulting in very low contributions to atmospheric pollution levels.







**Figure 5: Absolute contributions of local ship emissions to annual mean concentration levels in Gothenburg in 2012 (column 1) and BAU2040 (column 2), as well as the relative contributions (columns 3 and 4). © OpenStreetMap contributors 2019. Distributed under a Creative Commons BY-SA License.**





### 4.3 Influence of shore-side electricity use in future scenarios

The model simulations show that the contribution of shipping to air pollution in Gothenburg in the future scenarios is focussing on the port area (Fig. 5). The results for the shore-side electricity scenario BAU2040LP show visible reductions of $NO_2$, $PM_{2.5}$

and $SO_2$ concentrations in the port area (Fig. 6). For $NO_2$, local concentration reductions in the port area are up to 25 % in comparison to BAU2040 and for the EEDI2040LP scenario, the $NO_2$ reduction due to the shore-side electricity is up to 30 % (Fig. 6). In the surrounding areas of Gothenburg, the reductions range between 1 % and 15 %. In terms of $O_3$, replacement of emissions from auxiliary engines at berth with electricity (BAU2040LP scenario) causes an increase of up to 2.5 %. In the EEDI2040LP scenario the relative increase of annual mean $O_3$ concentrations compared to the EEDI2040 scenario is up to 3

% (Fig. 6). In both scenarios, the decrease in $NO_x$ emissions leads to an increase in $O_3$ concentrations due to less titration of $O_3$. The results for $PM_{2.5}$ and $SO_2$ show similar characteristics. BAU2040LP and EEDI2040LP lead to additional local reduction potentials of up to 3 % for $PM_{2.5}$ and 12 % for $SO_2$ in the port area but almost no difference can be seen outside the port area. EEDI2040LP shows a slightly higher reduction potential than BAU2040LP. In total, the implementation of shore-side electricity is clearly beneficial to reduce the impact of shipping emissions and therefore increase the air quality in areas

close to the port.

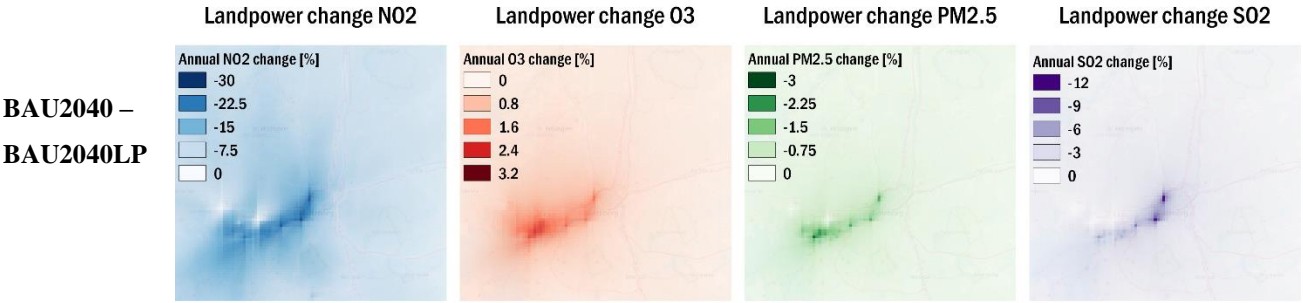

**Figure 6: Relative changes in annual mean $NO_2$, $PM_{2.5}$ and $O_3$ concentrations for BAU2040LP compared to 2040BAU scenario. © OpenStreetMap contributors 2019. Distributed under a Creative Commons BY-SA License.**

### 5. Impacts of future shipping on exposure to air pollutants and related health effects

### 5.1 Impact of future shipping on population exposure

Table 2 shows population weighted concentrations (PWC) for $NO_2$, $PM_{2.5}$ and $O_3$ (expressed as SOMO35) in the inner model domain for the year 2012 and the investigated scenarios BAU2040 and EEDI2040. PWC attributed to local and regional shipping in the Gothenburg area and the effect of shore-side electricity in BAU2040LP and EEDI2040LP are also shown. The calculated exposure to $PM_{2.5}$ was ~2.8 µg/m³ in the scenarios BAU2040 and EEDI2040 with large part of the exposure

originating from the regional background. The calculated decrease due to the reductions in anthropogenic emissions between



2012 and 2040 is by one third. The contribution of local shipping to $PM_{2.5}$ exposure was less than 1 % ($<0.02\ \mu g/m^3$) in all scenarios. In the BAU2040 scenario emissions from ships at berth which can be replaced by shore side electricity in the BAU2040LP scenario caused approximately 50 % of the $PM_{2.5}$ exposure attributed to the local shipping. In the EEDI2040LP scenario the shipping emissions at berth are responsible for more than 90 % of the exposure attributed to local shipping.

Although the BAU2040LP and EEDI2040LP scenarios imply a relative emission reduction of 62 % (compared to BAU2040 and EEDI2040 respectively) due to shore-side electricity use at berth, the impact in terms of absolute concentrations is less than 0.01 $\mu g/m^3$ in both scenarios. When additionally the contribution of emissions from regional shipping in the Baltic Sea and the North Sea is considered, the contribution of shipping to $PM_{2.5}$ exposure in the BAU2040 scenario is with 11 % (0.3 $\mu g/m^3$) substantially larger than that of the local shipping. This is due to the large impact of secondary particulate matter formed

during the atmospheric transport of the more distant emissions. This secondary PM is calculated in the CMAQ model providing the boundary conditions for the TAPM simulations and it considers mainly particulate sulphate and nitrate since VOC emissions from shipping are not included in that simulation.

The contribution of shipping to the PWC of $NO_2$ is about 10 % in the BAU2040 scenario and 20 % in the EEDI2040 scenario (0.1 and 0.3 ppb contributions, respectively). Thus, there is a clear impact of higher improved energy efficiency. When also

the contribution of regional shipping is considered in the BAU2040 scenario, the contribution of all shipping to the $NO_2$ exposure is 16 % (0.18 ppb), the regional shipping contribution is not as important as in the case of exposure to $PM_{2.5}$. The reason is that emissions of primary PM from shipping are by approximately 30 times lower than emissions of $NO_x$ and ~3 times lower than emissions of $SO_2$ which makes the contribution of secondary PM formed from the more distant emissions relatively more important compared to the local emissions of primary PM.

In all modelled scenarios the impact of local shipping is decreasing the population exposure to $O_3$ due to less titration in the absence of $NO_x$ sources. Nevertheless, when the impact of regional shipping is included in the BAU2040 scenario, shipping emissions cause small increases in exposure to $O_3$. The $O_3$ exposure attributed to local shipping considers, however, only population in the local region while one can expect $O_3$ formation causing an increase of exposure for the population living further away from the city.

In total, a very low impact of $PM_{2.5}$ due to local shipping activities was simulated for the scenarios with and without shore side electricity. In all cases, the regional background can be considered as the main contributor to $PM_{2.5}$ exposure in the urban area. For $NO_2$ the contribution of shipping related concentrations to the total air pollution is significant within both scenarios (with and without shore-side electricity reduction scenarios) in 2040, with important, but lower, contributions from regional background concentrations compared to $PM_{2.5}$. Nevertheless, the BAU2040LP and EEDI2040LP scenarios show a high

reduction potential and benefits for the air quality in densely located areas.



**Table 2: Population weighted exposure in the Gothenburg area to NO₂ (in ppb), PM₂.₅ (in µg/m³), and ozone (as sum of hourly means over 35ppb) in 2012 and in the BAU2040 and EEDI2040 scenarios. The exposure caused by local shipping, local and regional shipping and ships at berth is given separately.**

| Population weighted concentration (PWC) | NO₂ (ppb) | PM₂.₅ (µg/m³) | SOMO35 (ppb*h) |
|---|---|---|---|
| **Year 2012** | | | |
| Total base | 4.70 | 4.12 | 19698 |
| Local shipping * | 0.68 | 0.09 | -1186 |
| Local + regional shipping | 1.65 | 0.51 | -1115 |
| **Year 2040** | | | |
| Total BAU2040 | 1.16 | 2.80 | 18723 |
| Local shipping BAU2040 * | 0.08 | 0.02 | -115 |
| Local + regional shipping BAU2040 | 0.18 | 0.31 | 35 |
| Shipping emissions at berth** BAU2040 | 0.12 | 0.01 | -241 |
| Total EEDI2040 | 1.39 | 2.83 | 18434 |
| Local shipping EEDI2040 | 0.28 | 0.01 | -727 |
| Shipping emissions at berth** EEDI2040 | 0.18 | 0.01 | -267 |

\* Includes emissions at berth

\*\* Emissions avoided by being replaced by land-power in the BAU2040LP and EEDI2040LP scenarios

The contribution of air pollution levels to the overall population exposure expressed as PWC depends on the relationship between spatial distribution of concentrations of air pollutants and the population density. Fig. 7 shows products of mean concentration and population in each model grid cell with a resolution of 250 m x 250 m, which represent exposure in the

domain. Compared to the air quality situation in 2012 (Fig. 7a – 7c), the exposure to NO₂ and PM₂.₅ will decrease in the densely populated area in 2040, especially due to reduced emissions from road traffic. On the other hand, O₃ exposure is increasing correspondingly because of reduced local O₃ titration caused by decreased NO$_x$ emissions in most parts of the city. The spatial patterns of NO₂ and PM₂.₅ exposure from local shipping (Fig. 7g – 7i) are dominated by gradients in the concentration fields with highest reduction around the city ports north of the river Göta älv. The contribution of regional and local shipping to total

exposure (Fig. 7d – 7f) for NO₂ and PM₂.₅ are higher in a larger city area since regional-shipping-related NO₂ and PM₂.₅ exposure are evenly distributed over the city. The introduction of onshore electricity (Fig. 7j – 7 l) gives visible reductions in the port area for exposure to NO₂ (~ -200 ppb*capita) and PM₂.₅ (~ -50 µg/m³) due to the emissions avoided by shore-side electricity.





| PWC NO₂ (ppb*capita) | PWC PM₂.₅ (µg/m³*capita) | PWC SOMO35 (ppb*h*capita) |
|---|---|---|

(a)  (b)  (c)

**Total difference BAU2040 - 2012**

(d)  (e)  (f)

**Contribution of regional + local shipping in BAU2040**

(g)  (h)  (i)

**Contribution of local shipping in BAU 2040**

(j)  (k)  (l)

**Impact of shore-side electricity in BAU2040LP**





**Figure 7: Absolute contributions of total concentration changes between 2012 and 2040 to PWC in Gothenburg (a-c). Contribution of all shipping (regional + local) in BAU2040 to PWC (d-f). Contribution of local shipping in BAU2040 to PWC (g-i). Impact of implementation of shore-side electricity on PWC (j-l). © OpenStreetMap contributors 2019. Distributed under a Creative Commons BY-SA License.**

## 5.1 Impact of future shipping on health effects

Table 3 gives an overview of health impacts calculated for the exposure to $PM_{2.5}$, $NO_2$ and $O_3$ in terms of mortalities and years of life lost (YOLL). The results show that in 2040 all shipping-related $PM_{2.5}$, including the regional shipping, would cause 13 premature death/year corresponding to a shortened lifetime of 0.009 years per person (3.4 days) in the BAU2040 scenario. The majority of the impact (over 90 %) can be associated to the regional shipping outside the city. Impacts from local shipping in Gothenburg were found to be small, less than one premature death in the city, corresponding to 0.0006 and 0.0004 YOLLs/person in the BAU2040 and EEDI2040 scenario respectively (~0.2 days). Shore-side electricity reduced the impact of $PM_{2.5}$ from local shipping by ca. 40 % and 80 % in the BAU2040LP and EEDI2040LP scenarios, respectively, but only by few percent if also regional shipping is considered in the BAU2040 scenario. The impacts from the short-term exposure to $NO_2$ were calculated to be 0.36 premature deaths per year in the model domain for all shipping in the BAU2040 scenario, 0.17 premature death (46 %) being attributed to local shipping. In the EEDI2040 scenario the $NO_2$ impact from local shipping was larger, 0.55 premature deaths/year. Impacts from short-term exposure to $O_3$ associated to shipping were calculated to 0.02 premature deaths in the BAU2040 scenario when all shipping is considered. As emissions from the local shipping lead to a decrease in $O_3$ concentrations in the city, the impact of local shipping would decrease mortalities with 0.1 and 0.4 premature deaths per year in the BAU2040 and EEDI2040 scenarios, respectively.

The impact of climate policy measures in the shipping sector have been addressed in Cofala et al. (2018), too, in a study using similar methods as here. The study includes two base scenarios, one without and one with climate policy measures, the former keeping the shipping emissions by 45 % higher than the latter in 2040. Comparison of the data supplement in Cofala et al. (2018) shows that the $PM_{2.5}$-related mortalities caused by shipping decreased in the climate measures' scenario compared to the scenario without measures by 1.7 % (~300 YOLLs/year) for Sweden and by 2 % (~35'000 YOLLs/year) for the EU including UK, Norway and Switzerland. Mortalities caused by $O_3$ exposure related to shipping emissions decreased between these two scenarios by 4.3 % (6 premature deaths) for Sweden and by 5.4 % (848 premature deaths) for the EU in 2040. The results for $PM_{2.5}$ are in line with our findings and if we scale up our results to entire Sweden and account for the difference between the relative emission change in the scenarios by Cofala et al. (2018) and ours, we find rather similar result (~10 % difference). For $O_3$ the change between the scenarios in our study is of opposite sign than in Cofala et al. (2018) indicating difference in the $O_3$ formation regime in the two air pollution models used.



**Table 3: Health impacts calculated for BAU2040 and EEDI2040 scenarios with ARP and Rains methodologies: Emissions at berth - impacts potentially avoided by the shore-site electricity implementation (BAU2040 – BAU2040-LP and EEID2040 – EEID2040LP); Local shipping – impacts of all local shipping emissions in the model domain, including emissions at berth; Local + regional shipping – impacts of all shipping emissions including the regional shipping emissions in the Baltic Sea and the North Sea in the model boundary conditions; Total exposure – impacts of the total PWC for PM2.5, NO2 and ozone as SOMO35 in the model domain.**

| Pollutant | Impact | Unit | BAU2040 | | | | EEDI2040 | | |
| | | | Emissions at berth* | Local shipping** | Local + regional shipping | Total exposure | Emissions at berth* | Local shipping** | Total exposure |
|---|---|---|---|---|---|---|---|---|---|
| PM$_{2.5}$ | Chronic Mortality | Life years lost/year | 2.7 | 6.6 | 106 | 955 | 3.9 | 4.8 | 967 |
| PM$_{2.5}$ | Chronic Mortality | Premature deaths/year | 0.3 | 0.8 | 13 | 120 | 0.5 | 0.6 | 122 |
| PM$_{2.5}$ | Chronic mortality | YOLLs / person | 0.0002 | 0.0006 | 0.009 | 0.084 | 0.0003 | 0.0004 | 0.085 |
| PM$_{2.5}$ | Chronic Mortality (RAINS) | YOLLs / person | 0.0003 | 0.0007 | 0.011 | 0.096 | 0.0004 | 0.0005 | 0.097 |
| NO$_2$ | Acute Mortality | Premature deaths | 0.23 | 0.17 | 0.36 | 2.28 | 0.35 | 0.55 | 2.74 |
| O$_3$ | Acute Mortality | Premature deaths | -0.1 | -0.1 | 0.02 | 9.0 | -0.1 | -0.4 | 8.9 |

*Emissions at berth avoided by being replaced by land-power in the LP scenarios

** Includes emissions at berth

## 6 Conclusions

We investigated the future effect of shipping emissions on air quality and related health effects considering different scenarios of the development of shipping under current regional trends of economic growth and already decided regulations in the Gothenburg urban area in 2040. Additionally, we investigated the impact of a large-scale implementation of shore side electricity in the port of Gothenburg. For this purpose, we established a one-way nested chemistry transport modelling (CTM) system from the global to the urban scale, to calculate pollutant concentrations, population weighted concentrations and health-effects related to NO$_2$, PM$_{2.5}$ and O$_3$. This paper is the second part of a study about the current and future air quality situation in the Gothenburg urban area. Part 1 by Tang et al. (2020) introduced, evaluated and discussed air pollutant conccentraions, population weighted concentration and healt effects for the year 2012 and is published in the same special issue.

The simulated concentrations of NO$_2$ and PM$_{2.5}$ in future scenarios for the year 2040 are in general very low with up to 4 ppb for NO$_2$ and up to 3.5µg/m³ for PM$_{2.5}$ in the urban areas distant to the port area. Nevertheless, two hot spots of NO$_2$ and PM$_{2.5}$ with higher concentrations are located west of the city. These hotspots are due industrial emissions, in particular from refineries. Compared to 2012 the simulated overall exposure to PM2.5 decreased by approximately 30 % in the BAU scenario for 2040. For NO$_2$ the decrease was more than 60 %. The simulated concentrations of ozone increased between 2012 and 2040 by about 20 %. This increase results from both, higher concentrations simulated in the regional model domain and transported



through the boundaries into the city domain and from lower $O_3$ titration by reduced $NO_x$ emissions. Local shipping contributes also to the titration causing negative contributions to $O_3$ concentrations. The impact of the background shipping, however, causes $O_3$ formation in the city domain and the overall impact of shipping on $O_3$ concentrations is a net $O_3$ formation.

In general, the contributions of local shipping emissions in 2040 focus on the harbour area. Only to some extent, they influence

the rest of the city domain. For $NO_2$ there are in maximum local shipping contributions of more than 30 % to the total concentrations, located in the port and the area surrounding it, while contribution of the local shipping activities to the $NO_2$ concentrations in other parts of the city are up to 10 %, only. The contributions of $PM_{2.5}$ by shipping activities are generally much lower (up to 3 % at maximum) and follow the same trend with higher contributions located in the harbour area and its surroundings and lower impacts in the other areas.

The simulated impact of a wide use of shore-side electricity for shipping in 2040 shows similar spatial patterns for $NO_2$ and $PM_{2.5}$. For $NO_2$, local concentration reductions in the port area range between 25 % and 30 % at maximum, depending on the individual scenario. In the surrounding areas of Gothenburg, the reductions range between 1 % and 15 %. In terms of $O_3$, replacement of emissions from auxiliary engines at berth with electricity causes an increase of up to 2.5 % - 3 %, depending on the scenario. Implementation of shore side electricity for ships at berth leads to an additional local reduction potential of up

to 3 % for $PM_{2.5}$ and 12 % for $SO_2$ in the port area but almost no difference can be seen outside the port area. In total, the implementation of shore-side electricity is clearly beneficial to reduce the impact of shipping emissions and therefore improves air quality in areas close to the port. Moreover, the strict regulations as simulated in the BAU2040 scenario are of high value for an improved air quality in the urban area of Gothenburg.

Calculated population weighted concentrations and health impacts follow the same trends. The $PM_{2.5}$ contribution of local

shipping to PWC of $PM_{2.5}$ is below 1 % for all scenarios, but the contribution of regional shipping outside the city domain is still of importance: In the BAU2040 scenario, its contribution is about 10 %. Relative to 2012 the exposure to $PM_{2.5}$ from local shipping decreased by more than 80 % in all scenarios for 2040 while the impact of all shipping including the regional contribution decreased by only 40 %. The local shipping contribution to PWC of $NO_2$ is much more pronounced, being 10 % for the BAU2040 and 20 % for the EEDI2040 scenario. In the BAU scenario, regional shipping contributed additional 6 % to

the PWC of $NO_2$. In comparison to 2012, exposure to $NO_2$ from local shipping decreased by approximately 50 % for the EEDI2040 scenario and by more than 80 % for the BAU2040 scenario. In the BAU2040 scenario exposure to $NO_2$ from all shipping, including the regional contribution, decreased by 86 %. The PWC of ozone, given as SOMO35, is increasing in the model domain by about 20 %, following the trends in concentrations.

The most serious health effects were associated to $PM_{2.5}$. It needs to be emphasized that the effects presented in Table 3 cannot

be added due to a risk of double-counting, especially concerning $PM_{2.5}$ and $NO_2$. Between 2012 and 2040 we can see a large decrease in mortality caused by $PM_{2.5}$ associated with shipping, in the future reference scenario (BAU2040) the total decrease is 86 %. For the whole city-domain, this results in 9 premature deaths per year that are avoided, corresponding to 67 YOLLs/year. This decrease is mainly associated to the decrease in emissions due to strengthened SECA legislation introduced in 2015. The introduction of the NECA legislation in 2021, with a rather slow uptake of abatement technologies for the





reduction of $NO_x$ emissions, as well as climate policy measures implemented in terms of energy effectivization of the fleet, will lead to an additional reduction of $CO_2$ emissions and those of other air pollutants. Partial impacts of these three aspects have not been studied here.

As already discussed in Tang et al. (2020), Jonson et al. (2019) studied the impact of an introduction of strengthened sulphur limits in 2015 and found approximately 35 % reduction of the impact from the regional shipping contribution to $PM_{2.5}$ around Gothenburg. The global study of Sofiev et al. (2018) shows that the impact of the global cap down to 0.5 % FSC does not have any significant impact on a further reduction of shipping-related air pollution around the Swedish West coast. The calculated decrease of $PM_{2.5}$ was below 1 %. Only limited impacts of these two regulations on emission from the local shipping in Gothenburg can be expected since ships at berth have been using fuels with maximum FSC 0.1 % since 2010, already.

In this study, the impact of the improved energy efficiency can be obtained by comparing the BAU scenario with the EEDI scenario, of the latter showing approx. 30 % higher emissions. The impact on health effects from exposure to $PM_{2.5}$ in the city domain is rather low, 1.2 % only. The impact of the higher emissions on exposure to NO2 is much more important with 17 %, while exposure to ozone is lower in the EEDI scenario by 1.6 %, compared to BAU.

Nevertheless, when it comes to the applied static exposure approach to calculate PWC and health effects, the underlying assumption is that people are at residential addresses throughout the time. Thus, static exposure does not account for spatial and temporal variability of population activities and will lead to uncertainties in calculated exposure and introduce potential bias in the quantification of human health effects. Several exposure modelling studies have overcome this traditional approach and are using population activity data, derived from surveys, individual GIS data or generic data, and models to account for the diurnal variation in population numbers in different locations (e.g. Beckx et al., 2009; Smith et al., 2016; Ramacher et al., 2019b; Ramacher and Karl, 2020; Reis et al., 2018; Soares et al., 2014). Thus, to model population numbers suitable for exposure calculations, it is generally necessary to know the population distribution and characterization and therefore the number of people and diurnal activity patterns of different characteristic population groups. In future studies we plan to account for population dynamics by applying averaged generic population activity profiles, which are additionally diversified by demographic groups in different microenvironments, such as residential environments, work environments or traffic environments. This will allow for a better representation of pollutant concentrations people are exposed to and the related health effects that are based on exposure calculations.

Impacts of the local shipping emissions on air quality and human health are further discussed and evaluated for the year 2012 in the part I paper (Tang et al., 2020).

This article is part of the special issue "Shipping and the Environment 2017"

**Code availability**

The TAPM model is a commercial software available at CSIRO, Australia (www.csiro.au). STEAM model is intellectual property of the Finnish Meteorological Institute and is not publicly available. ARP is commercial software available from arirabl.com.

**Data availability**

The model output data are available upon request from the corresponding authors

**Author contribution**

MOPR, LT, JM and VM designed the model simulations. LJ and JPJ calculated ship emissions with the STEAM model and contributed with text about the shipping emissions, LT prepared ship emission files for the model simulations. EF prepared
future scenarios. LT, MG and JM prepared emission data from other sources. MK prepared data from the regional-scale simulation used for the boundary. MOPR and LT prepared the model set-up and other input data, performed the model simulations and evaluated the model results. LT calculated exposures and JM and KY calculated the health impacts. MOPR and JM wrote the major part of the text with assistance from LT and VM.

**Competing interests**

JM is associated editor of the special issue Shipping and Environment.

**Acknowledgements**

The air quality model CMAQ is developed and maintained by the U.S. Environmental Protection Agency (US EPA). COSMO-CLM is the community model of the German climate research (www.clm-community.eu). The simulations with COSMO-CLM and CMAQ were performed at the German Climate Computing Centre (DKRZ) within the project "Regional
Atmospheric Modelling" (Project Id 0302). The Swedish Meteorological and Hydrological Institute (SMHI) is thanked for making available the precipitation data from rain gauge stations in Sweden. Z. Klimont (IIASA) is thanked for emission data for the 2040 CLE scenario from ECLIPSE v5. NILU is thanked for the EBAS database maintenance and data provision. Sara Jutterström (IVL) is thanked for good collaboration and discussion of model results on deposition of nitrogen and sulphur.



**Financial support**

This work has been conducted within the BONUS SHEBA (Sustainable Shipping and Environment of the Baltic Sea region) research project under Call 2014-41. BONUS (Art 185), funded jointly by the EU, Swedish Environmental Protection Agency, Academy of Finland and by the German Federal Ministry of Education and Research under Grant Number 03F0720A and within project platform CSHIPP, subsidy contract #C006 of Interreg Baltic Sea Region.

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

    rational%20Measures/Resolution%20MEPC.203(62).pdf, last access: 25 March 2020.



IMO: Amendments to the Annex of the Protocol of 1997 to Amend the International Convention for the Prevention of Pollution from Ships, 1973, as Modified by the Protocol of 1978 Relating thereto. (Amendments to regulations 2, 13, 19, 20 and 21 and the Supplement to the IAPP Certificate under MARPOL Annex VI and certification of dual-fuel engines under the NOX Technical Code 2008): Resolution MEPC 251(66):

http://www.imo.org/en/KnowledgeCentre/IndexofIMOResolutions/Marine-Environment-Protection-Committee-%28MEPC%29/Documents/MEPC.251%2866%29.pdf, last access: 25 March 2020.

IMO: Amendments to the Annex of the Protocol of 1997 to Amend the International Convention for the Prevention of Pollution from Ships, 1973, as Modified by the Protocol of 1978 Relating thereto. (Designation of the Baltic Sea and the North Sea Emission Control Areas for NOX Tier III control): Resolution MEPC 286(71):

http://www.imo.org/en/KnowledgeCentre/IndexofIMOResolutions/Marine-Environment-Protection-Committee-%28MEPC%29/Documents/MEPC.286%2871%29.pdf, last access: 25 March 2020.

IMO: Initial IMO strategy on reduction of GHG emissions from ships: Resolution MEPC 304(72): http://www.imo.org/en/KnowledgeCentre/IndexofIMOResolutions/Marine-Environment-Protection-Committee-%28MEPC%29/Documents/MEPC.304%2872%29.pdf, last access: 25 March 2020.

Jalkanen, J.-P., Brink, A., Kalli, J., Pettersson, H., Kukkonen, J., and Stipa, T.: A modelling system for the exhaust emissions of marine traffic and its application in the Baltic Sea area, Atmos. Chem. Phys., 9, 9209–9223, doi:10.5194/acp-9-9209-2009, 2009.

Jalkanen, J.-P., Johansson, L., Kukkonen, J., Brink, A., Kalli, J., and Stipa, T.: Extension of an assessment model of ship traffic exhaust emissions for particulate matter and carbon monoxide, Atmos. Chem. Phys., 12, 2641–2659,

doi:10.5194/acp-12-2641-2012, 2012.

Johansson, L., Jalkanen, J.-P., Kalli, J., and Kukkonen, J.: The evolution of shipping emissions and the costs of regulation changes in the northern EU area, Atmos. Chem. Phys., 13, 11375–11389, doi:10.5194/acp-13-11375-2013, 2013.

Jonson, J. E., Gauss, M., Jalkanen, J.-P., and Johansson, L.: Effects of strengthening the Baltic Sea ECA regulations, Atmos. Chem. Phys., 19, 13469–13487, doi:10.5194/acp-19-13469-2019, 2019.

Jonson, J. E., Jalkanen, J. P., Johansson, L., Gauss, M., and van der Denier Gon, H. A. C.: Model calculations of the effects of present and future emissions of air pollutants from shipping in the Baltic Sea and the North Sea, Atmos. Chem. Phys., 15, 783–798, doi:10.5194/acp-15-783-2015, 2015.

Kalli, J., Jalkanen, J.-P., Johansson, L., and Repka, S.: Atmospheric emissions of European SECA shipping: long-term projections, WMU J Marit Affairs, 12, 129–145, doi:10.1007/s13437-013-0050-9, 2013.

Karl, M., Bieser, J., Geyer, B., Matthias, V., Jalkanen, J.-P., Johansson, L., and Fridell, E.: Impact of a nitrogen emission control area (NECA) on the future air quality and nitrogen deposition to seawater in the Baltic Sea region, Atmos. Chem. Phys., 19, 1721–1752, doi:10.5194/acp-19-1721-2019, 2019a.





Karl, M., Jonson, J. E., Uppstu, A., Aulinger, A., Prank, M., Sofiev, M., Jalkanen, J.-P., Johansson, L., Quante, M., and Matthias, V.: Effects of ship emissions on air quality in the Baltic Sea region simulated with three different chemistry transport models, Atmos. Chem. Phys., 19, 7019–7053, doi:10.5194/acp-19-7019-2019, 2019b.

Keller, M., Hausberger, S., Matzer, C., Wüthrich, P., and Notter, B.: HBEFA Version 3.3: Background Documentation: http://www.hbefa.net/e/documents/HBEFA33_Documentation_20170425.pdf, last access: 8 January 2020.

Kiesewetter, G., Borken-Kleefeld, J., Schöpp, W., Heyes, C., Thunis, P., Bessagnet, B., Terrenoire, E., Gsella, A., and Amann, M.: Modelling $NO_2$ concentrations at the street level in the GAINS integrated assessment model: projections under current legislation, Atmos. Chem. Phys., 14, 813–829, doi:10.5194/acp-14-813-2014, 2014.

Matthaios, V. N., Triantafyllou, A. G., Albanis, T. A., Sakkas, V., and Garas, S.: Performance and evaluation of a coupled prognostic model TAPM over a mountainous complex terrain industrial area, Theor Appl Climatol, 132, 885–903, doi:10.1007/s00704-017-2122-9, 2018.

Ramacher, M. O. P. and Karl, M.: Integrating Modes of Transport in a Dynamic Modelling Approach to Evaluate Population Exposure to Ambient NO2 and PM2.5 Pollution in Urban Areas, IJERPH, 17, 2099, doi:10.3390/ijerph17062099, 2020.

Ramacher, M. O. P., Karl, M., Aulinger, A., and Bieser, J.: Population Exposure to Emissions from Industry, Traffic, Shipping and Residential Heating in the Urban Area of Hamburg, in: Air Pollution Modeling and its Application XXVI, Mensink, C., Gong, W., and Hakami, A. (Eds.), Springer Proceedings in Complexity, Springer International Publishing, Cham, 177–183, 2020.

Ramacher, M. O. P., Karl, M., Bieser, J., Jalkanen, J.-P., and Johansson, L.: Urban population exposure to $NO_x$ emissions from local shipping in three Baltic Sea harbour cities – a generic approach, Atmos. Chem. Phys. Discuss., 1–45, doi:10.5194/acp-2019-127, 2019a.

Ramacher, M. O. P., Karl, M., Bieser, J., Jalkanen, J.-P., and Johansson, L.: Urban population exposure to NOx emissions from local shipping in three Baltic Sea harbour cities – a generic approach, Atmos. Chem. Phys., 19, 9153–9179, doi:10.5194/acp-19-9153-2019, 2019b.

Reis, S., Liška, T., Vieno, M., Carnell, E. J., Beck, R., Clemens, T., Dragosits, U., Tomlinson, S. J., Leaver, D., and Heal, M. R.: The influence of residential and workday population mobility on exposure to air pollution in the UK, Environment international, 121, 803–813, doi:10.1016/j.envint.2018.10.005, 2018.

Rockel, B., Will, A., and Hense, A.: The Regional Climate Model COSMO-CLM (CCLM), metz, 17, 347–348, doi:10.1127/0941-2948/2008/0309, 2008.

Sillman, S.: The relation between ozone, NOx and hydrocarbons in urban and polluted rural environments, Atmospheric Environment, 33, 1821–1845, doi:10.1016/S1352-2310(98)00345-8, 1999.

Smith, J. D., Mitsakou, C., Kitwiroon, N., Barratt, B. M., Walton, H. A., Taylor, J. G., Anderson, H. R., Kelly, F. J., and Beevers, S. D.: London Hybrid Exposure Model: Improving Human Exposure Estimates to NO2 and PM2.5 in an Urban Setting, Environmental science & technology, 50, 11760–11768, doi:10.1021/acs.est.6b01817, 2016.





Soares, J., Kousa, A., Kukkonen, J., Matilainen, L., Kangas, L., Kauhaniemi, M., Riikonen, K., Jalkanen, J.-P., Rasila, T., Hänninen, O., Koskentalo, T., Aarnio, M., Hendriks, C., and Karppinen, A.: Refinement of a model for evaluating the population exposure in an urban area, Geosci. Model Dev., 7, 1855–1872, doi:10.5194/gmd-7-1855-2014, 2014.

Sofiev, M., Winebrake, J. J., Johansson, L., Carr, E. W., Prank, M., Soares, J., Vira, J., Kouznetsov, R., Jalkanen, J.-P., and
Corbett, J. J.: Cleaner fuels for ships provide public health benefits with climate tradeoffs, Nature communications, 9, 406, doi:10.1038/s41467-017-02774-9, 2018.

Tang, L., Ramacher, M. O. P., Moldanova, J., Matthias, V., Karl, M., and Johansson, L.: The impact of ship emissions on air quality and human health in the Gothenburg area – Part 1: Current situtation, Atmos. Chem. Phys. Discuss., in preparation, 2020.

Transport administration: Prognos för persontrafiken 2040: Trafikverkets Basprognoser 2016, Report 2016:062, Transport administration (Trafikverket), 2016.

Transport administration: Prognos för persontrafiken 2040: Trafikverkets Basprognoser 2018-04-01, Report 2018:089, Transport administration (Trafikverket), 2018.

Zandersen, M., Hyytiäinen, K., Meier, H. E. M., Tomczak, M. T., Bauer, B., Haapasaari, P. E., Olesen, J. E., Gustafsson, B.
G., Refsgaard, J. C., Fridell, E., Pihlainen, S., Le Tissier, M. D. A., Kosenius, A.-K., and van Vuuren, D. P.: Shared socio-economic pathways extended for the Baltic Sea: exploring long-term environmental problems, Reg Environ Change, 19, 1073–1086, doi:10.1007/s10113-018-1453-0, 2019.