# Peer review of "The impact of ship emissions on air quality and human health in the Gothenburg area – Part II: Scenarios for 2040"

_Atmospheric Chemistry and Physics, 2020_

## Referee Comment (RC1) · Anonymous Referee #1 · 12 May 2020

This study predicted the impact of ship emissions on air quality and human health in the Gothenburg area. Air quality simulations for four future scenarios were conducted to evaluate impacts of emission changes from 2012 to 2040, local shipping activities in 2040, and local port measures.

Major comments:

1. I can understand that the impacts of following items can be evaluated in this study as described in the lines 7-9 of the page 13.

(1) the impact on air quality in Gothenburg through a change in total emissions from 2012 to 2040 (2) the impact of local shipping activities in 2040 in two different scenarios

(3) the additional impact of local port measures (shore-side electricity) in scenarios for 2040

But, how can these results be utilized? For example, if predicted improvement of air quality is compared with the air quality standards or any targets, you can say that existing emission regulations for ships are enough or not. Health impacts were calculated in this study. They can also be compared with any targets to judge if existing regulations are enough to suppress health impacts below the certain target. However, such a judgement was not conducted in this study. Number of scenarios are very limited and do not include any additional emission control measures for ships. In the current design, it is also difficult to judge if local port measures are necessary or not. Please add more explanations on utilizing the results of this study.

2. As mentioned in the lines 17-19 of the page 2, in combination with the increasing ship traffic which grows roughly by 2 % per year and the future foreseeable significant decrease of emissions from other anthropogenic sectors, the relative importance of NOx emissions from shipping for urban air quality will thus likely remain high. In addition, as mentioned in the lines 6-7 of the page 4, scenarios for transported cargo volumes, composition of the fleet, as well as energy efficiency improvements need to be developed and put into perspective with probable emission reductions at land. Then, how were future changes in ship traffics, cargo volumes, and fleet compositions treated in this study? How are emission increases by them compensated by reductions by regulations?

3. While overall descriptions of Gothenburg should be included in the Part I paper, some readers may not know where Gothenburg is and how it looks like. Simple descriptions may be helpful. In addition, all the contours do not show any geographical and administrative boundaries. They may be also helpful to recognize where land, ocean, and ports are.

Specific comments:

P1, L27-30 Two expressions, "wide use of shore-site electricity" and "implementation of on-shore electricity", are confusing. They correspond to the same thing, right?

P3, L4-5 Critical not only for NO2 and O3 but also PM2.5 as mentioned above.

P3, L10 Where is "this region"? It is not explicitly mentioned in preceding sentences.

P5, Figure 1 This figure is not mentioned in any sentence in the manuscript.

P5, L4 What is the reason to couple TAPM and CMAQ instead of using TAPM or CMAQ only? Is that described in the Part I paper?

P5, L16 I think that evaluations cannot be done for the future year 2040.

P6, L9-14 Is it appropriate to use different horizontal resolutions for air pollution and meteorological fields? How to interpolate or extrapolate either of them?

P7, L22-23 I suppose that a scaling factor for combustion in industry for energy purposes is large because their VOC emissions are very low. Please check their emission amounts in GAINS. But that is not in the case in the emissions used in this study shown in Figure 2. It might be due to inconsistent definitions of sectors. Is there any appropriate reason explaining why their emissions significantly increase in Gothenburg?

P12, L5-6 I cannot understand differences between BAU2040 and EEDI2040 for fuel efficiencies. What kind of policies are assumed in each scenario? Why fuel efficiency is higher in BAU2040 than EEDI2040? What is the motivation to compare these two scenarios? Please add more explanations to clarify significance of EEDI2040 scenario.

P13, 3.3 Scenario setup Were simulations conducted for twelve months in 2012?

P15, L4-5 Is 7% reduction a high potential? In fact, I cannot understand which part of Figure 4 this sentence describes.

P15, L18-20 I cannot believe such high PM2.5 concentrations according to Figure 4. In addition, I think the units of absolute and relative differences of PM2.5 in Figure 4 are

opposite.

P17, L4 It is confusing to represent changes in negative contributions to O3 as "increasing".

P20, L16-19 I think longer lifetime of secondary components in the atmosphere should be also one of reasons.

References Some references have no years. Particularly, it is difficult to distinguish IMO reports.

---

## Referee Comment (RC2) · Anonymous Referee #2 · 14 Jun 2020

This study conducted simulations to assess the impact of ship emissions on air quality over the Gothenburg area, as well as their health impacts between 2012 and 2040. The manuscript is well written and organized. I recommend this manuscript to be published if the comments are addressed.

Major comments:

1. Please add general descriptions of Gothenburg, including its graphical locations. Moreover, please add longitudes, latitudes and geographical information for all the spatial distribution maps in the manuscript.

2. This study adopted meteorological field of 2012 in the simulation. The diffusion con-

ditions may influence the impacts of emission reduction on air quality. So please add descriptions of the meteorological fields of 2012 to describe whether it is a year with good diffusion conditions or not. I suggest selecting a year of which the meteorological conditions are close to the climatological conditions, and then conduct the simulation.

3. In Section 5, this study assessed the impact of future shipping on human health, including premature deaths because the decrease of ambient PM2.5, O3, and NO2. Exposures to PM2.5, NO2, and O3 can all lead to premature deaths due to respiratory diseases. So in Table3, I am wondering whether there are overlaps between the number of premature deaths due to PM2.5 with those due to NO2 and O3.

Minor comments: 1. P6 Line10-15: Add more information for the simulation, including a figure to present the domains of the simulation, the period of the simulation, model spin-up, etc. 2. Cite Figure 1 in the manuscript, or delete it. 3. Please show the spatial distribution of the emission inventories of 2012. 4. In Figure 4, the unit for figure in row3 column 3 should be ug/m3; the unit for figure in row3 column 4 should be %.

---

## Author Response (AR1)

**Response to reviewer comment #1**

https://doi.org/10.5194/acp-2020-319-RC1, 2020

Ref.: acp-2020-319
5   https://doi.org/10.5194/acp-2020-319-RC2, 2020
Title: The impact of ship emissions on air quality and human health in the Gothenburg area – Part II
Authors: Martin O. P. Ramacher, Lin Tang, Jana Moldanová, Volker Matthias, Matthias Karl, Erik Fridell, and
Lasse Johansson
Journal: Atmospheric Chemistry and Physics, Special Issue Shipping and the Environment – From Regional to
10   Global Perspectives (ACP/OS inter-journal SI)

**Point 1:** This study predicted the impact of ship emissions on air quality and human health in the Gothenburg area. Air quality
simulations for four future scenarios were conducted to evaluate impacts of emission changes from 2012 to 2040, local shipping
activities in 2040, and local port measures.

15   **Response to point 1:** We thank the Reviewer for providing a detailed evaluation of our study, the manuscript and the helpful
comments and suggestions regarding the methodology used in our study.

**Point 2:** I can understand that the impacts of following items can be evaluated in this study as described in the lines 7-9 of the
page 13. (1) the impact on air quality in Gothenburg through a change in total emissions from 2012 to 2040 (2) the impact of
20   local shipping activities in 2040 in two different scenarios (3) the additional impact of local port measures (shore-side
electricity) in scenarios for 2040. But, how can these results be utilized? For example, if predicted improvement of air quality
is compared with the air quality standards or any targets, you can say that existing emission regulations for ships are enough
or not. Health impacts were calculated in this study. They can also be compared with any targets to judge if existing regulations
are enough to suppress health impacts below the certain target. However, such a judgement was not conducted in this study.

25   **Response to point 2:** We thank the reviewer for pointing out the need to compare our results to existing standards or targets
for a better utilization of this study. To our knowledge, there are no direct targets in terms of acceptable levels of health impacts
from air pollution, such as shortened lifetime, as it is politically a sensitive issue. Instead, the limit values and especially the
target values included in the EU Air Quality Directive (AQD, Directive 2008/50/EU) and WHO Air Quality Guidelines (AQG)
are based on health impact assessment, cost-benefit analysis and extensive discussion on this issue. The comparison shows
30   that simulated concentrations in Gothenburg in 2040 are below annual and hourly AQD and AGQ target, limit and guideline
values for all criteria pollutants. Annual analyses of air quality monitoring data Environmental Administration of City of
Gothenburg show exceedances of both the target and the limit values for $NO_2$ at several stations in Gothenburg in 2012 with
decreasing trends towards exceedances of only the limit value at traffic stations in 2019. For $PM_{10}$ the levels were well below
the limit value but exceeding the target value in 2012 without any significant trend towards presence with exception of the

urban background where slightly decreasing trend have been observed and the annual mean was bellow the target value of 15 µg/m³ the last 4 years. The measured concentration levels of $PM_{2.5}$ have been bellow the target value without any significant trend at Gothenburg monitoring stations. Concentrations of ozone have a slightly increasing trend from year 2012 onwards and tend to exceed the limit values for maximum hourly and 8-h means at a number of occasions each year (Miljöförvaltningen,

5  2019).

In the accompanying publication (Part 1, Tang et al., 2020, accepted & in production) the pollutant concentrations in 2012 have been simulated, compared to measurements and analyzed in detail. In our simulations for 2012, there are only few exceedances of the hourly $NO_2$ AQD limit and AQG guideline value (200 µg/m³) close to road traffic. When it comes to $O_3$, there are multiple exceedances of the maximum-daily-8-hour-mean (MDA8) AQD target value of 120 µg/m3 and even more

10  for the AQG guideline value of 100 µg/m³ measured and modelled in 2012. Similar holds true for $PM_{10}$ and $PM_{2.5}$: while there are no exceedances of annual target or limit values, some exceedances of the 24 hour mean target and guideline values for $PM_{10}$ (50 µg/m³) are measured in 2012, some exceedances for the $PM_{2.5}$ target value (25 µg/m³), but more for the $PM_{2.5}$ guideline value of 10 µg/m³.

Such exceedances have been avoided in all 2040 scenarios. Nevertheless, reports on health impacts from PM suggest that there

15  is no threshold PM concentration below which the exposure would not lead to adverse health effects. Thus, it is necessary to keep PM concentrations as low as possible, especially close to residential areas. Following, when it comes to harbor cities it is necessary to keep shipping related emissions at a minimum as their contribution is in relation to other sources important also in the future. Our scenario results for 2040 showed that the different shipping scenarios lead to decreases of PM concentrations. We added this information in the manuscript in the text passages summarized hereunder:

20  New section 2.1 on Gothenburg: "Annual analyses of air quality monitoring data Environmental Administration of City of Gothenburg show exceedances of both the target and the limit values for $NO_2$ at several stations in Gothenburg in 2012 with decreasing trends towards exceedances of only the limit value at traffic stations in 2019. For $PM_{10}$ the levels were well below the limit value but exceeding the target value in 2012 without any significant trend towards presence with exception of the urban background where slightly decreasing trend have been observed and the annual mean was below the target value of 15

25  µg/m³ the last 4 years. The measured concentration levels of $PM_{2.5}$ have been below the target value without any significant trend at Gothenburg monitoring stations. Concentrations of ozone have a slightly increasing trend from year 2012 onwards and tend to exceed the limit values for maximum hourly and 8-h means at a number of occasions each year (Miljöförvaltningen, 2019 )."

Results section: "In 2040 simulations the air quality situation in the city improves considerably and the concentrations will be

30  below the air quality limit and target values both for $NO_2$ and $PM_{2.5}$ in the city, even if the potential underestimates of the model system is accounted for (Tang et al., 2020, accepted & in production). From the static point of view of year 2040 conclusion that additional measures to reduce air pollution levels beyond those in the BAU scenario would not be necessary could be drawn. However, in perspective taking also the temporal development into consideration, the measures reducing concentrations of $NO_2$ will be implemented only in a slow pace and full impact of many of them, especially those targeting

shipping, will first be seen in the time horizon of 2040. The local measures could be, on the other hand, implemented faster and the significant reduction of $NO_2$ concentrations from implementation of on-shore electricity could thus reduce the time before the air quality targets are met in the city."

5 **Point 3:** Number of scenarios are very limited and do not include any additional emission control measures for ships. In the current design, it is also difficult to judge if local port measures are necessary or not. Please add more explanations on utilizing the results of this study.

**Response to point 3:** We thank the reviewer for critically examining the descriptions of the applied scenarios and agree on the need to further elaborate on the differences and details of the scenarios.

10 This study has been conducted within the BONUS SHEBA project (Shipping and Environment of the Baltic Sea Region) where the impact of current and scenario emissions from ships on air quality have been investigated as a part of a holistic assessment framework for impacts of shipping on marine and coastal environment. Much larger number of shipping scenarios have been investigated in terms of drivers of the shipping and emissions to air and seawater (Fridell et al., 2015). Selected number of scenarios regarding the shipping-related air pollution have been investigated on a range of spatial scales with several

15 chemistry-transport models: coarse spatial scale resolution was used for simulations in the European domain, finer resolution was used for the Baltic Sea (Karl et al., 2019b; Karl et al., 2019a), and city-scale simulations using high spatial resolution were used for several harbour cities (Ramacher et al., 2019). Part 1 of this study evaluates the contributions of regional and local shipping to the concentrations of $SO_2$, $NO_2$, $PM_{2.5}$, $O_3$ and secondary PM, as well as the human exposure and the associated health impacts in Gothenburg for year 2012.

20 All scenarios developed in the course SHEBA aim at identifying the impacts of existing and realistic regulations for shipping, which are decided but not yet in force. Taking into account reasonable projections of land-based emission inventories, as well as background pollutant concentrations, which have been derived in regional-scale studies (Karl et al. 2019) following the same scenarios of shipping and land-based emissions, the scenarios can be considered as realistic projections of future conditions. The underlying assumptions in the scenarios are described in detail in Fridell et al. (2015) and are translated to

25 emission scenarios in Karl et al. (2019) as described and referenced in our manuscript. For clarification of the need for the EEDI scenario, we added the following to the scenario description:

"As the scenario work revealed that energy effectivization has large impact on emissions in the target year, encompassing at the same time large uncertainty, we have chosen to include an alternative scenario with different effectivization level."

While the underlying assumptions that have led to the scenario descriptions have an impact on the whole Baltic Sea region,

30 we have decided to additionally analyze the impact of on-shore electricity in Gothenburg, which can be considered as a feasible local emission reduction strategy. Thus, the underlying assumptions and regulations in the BAU2040 and EEDI2040 scenario cannot be influenced by local authorities in Gothenburg, while local measures such as on-shore electricity can be applied by local authorities. Our results show, that especially for $NO_2$, the installation of on-shore electricity can lead to substantial reductions of pollutant concentrations and therefore should be implemented as soon as possible, accompanying regional

regulations, especially taking the fact that the city has currently problem with exceedances of the limit and target values for $NO_2$ concentrations into the consideration. In our future scenarios the air quality situation in the city improves considerably and one can see that the concentrations are bellow the target values both for $NO_2$ and $PM_{2.5}$, so from the static point of view of year 2040 additional measures to reduce air pollution levels beyond those in the BAU scenario would not be necessary. In perspective taking also the temporal development into consideration the measures reducing concentrations of $NO_2$ will be implemented only in a slow pace and full impact of many of them, especially those targeting shipping, will first be seen in the time horizon of 2040. The local measures could be implemented faster and the significant reduction of $NO_2$ concentrations from implementation of on-shore electricity could thus reduce the time before the air quality targets are met in the city. We have added this reasoning to conclusions of the study:

"In 2040 simulations the air quality situation in the city improves considerably and the concentrations will be below the air quality limit and target values both for $NO_2$ and $PM_{2.5}$ in the city, even if the potential underestimates of the model system is accounted for (Tang et al., 2020). From the static point of view of year 2040 conclusion that additional measures to reduce air pollution levels beyond those in the BAU scenario would not be necessary could be drawn. However, in perspective taking also the temporal development into consideration, the measures reducing concentrations of $NO_2$ will be implemented only in a slow pace and full impact of many of them, especially those targeting shipping, will first be seen in the time horizon of 2040. The local measures could be, on the other hand, implemented faster and the significant reduction of $NO_2$ concentrations from implementation of on-shore electricity could thus reduce the time before the air quality targets are met in the city."

**Point 4:** As mentioned in the lines 17-19 of the page 2, in combination with the increasing ship traffic which grows roughly by 2 % per year and the future foreseeable significant decrease of emissions from other anthropogenic sectors, the relative importance of NOx emissions from shipping for urban air quality will thus likely remain high. In addition, as mentioned in the lines 6-7 of the page 4, scenarios for transported cargo volumes, composition of the fleet, as well as energy efficiency improvements need to be developed and put into perspective with probable emission reductions at land. Then, how were future changes in ship traffics, cargo volumes, and fleet compositions treated in this study? How are emission increases by them compensated by reductions by regulations?

**Response to point 4:** We thank the reviewer for pointing out the need for a better description of our scenarios. We realized that the information on fleet development was not given in the paper and therefore added the following passage:

"The ship traffic volumes are expected to continue to grow with about $1\%\,yr^{-1}$ on average (it varies with ship type); the current trend of using larger vessels is expected to continue as well (Kalli et al. (2013). The trends in cargo volumes, passenger numbers and ship-sizes are in detail described in Fridell et al., 2015 and translated to emission scenarios in Karl et al. (2019)." The impact of legislation comparing to the other trends can be observed when comparing the different scenarios with the difference between BAU2040 and 2012: EEDI and BAU shows impact of energy effectivization, scenarios with and without on-shore electricity show impact of Karl et al. (2020) shows impact of the NECA legislation.

Comparing the emission totals for local shipping in Gothenburg show that in BAU2040 scenario emissions dropped comparing to 2012 by 91%, 85%, 78% and 31% for $SO_2$, PM, $NO_x$ and NMHC emissions, respectively, as a result of regulations of $SO_2$ and $NO_x$ emissions and energy effectivization under the scenario prediction of growth of the shipping sector. The EEDI2040LP scenario which implements energy effectivization exactly as described by the legislation show emission decrease relative to 2012 by 88%, 77%, 68% and 2% for $SO_2$, PM, $NO_x$ and NMHC emissions, respectively. This means that for emitted species that are targeted by legislation regulating emissions of air pollutants, i.e. $SO_2$ and $NO_x$, the lower energy effectivization led to a scenario with emissions reduction of 96% and 92% of that in BAU, a relatively small difference. Also for primary PM emissions, which are significantly affected by both abatement, measures the lower energy effectivization led to a scenario with emissions reduction rather close to BAU, 88% of that in BAU. VOC emissions are, on the other hand, not largely affected by the air pollution abatement measures and while the BAU scenario shows 31% reduction, in the EEDI scenario NMHC emissions are close to 2012 emissions, i.e. the energy effectivization.

Comparing emissions in BAU2040 and BAU2040LP scenarios or in EEDI2040 and EEDI2040LP scenarios gives us information about influence of hoteling emissions that can be replaced by on-shore electricity in these scenarios. Since proportion between hoteling and total emissions is the same in BAU and EEDI, the potential for relative reduction of emissions is also the same for both: 37% for $SO_2$, 64% for $NO_x$, 60% for PM and 68% for NMHC.

**Point 5:** While overall descriptions of Gothenburg should be included in the Part I paper, some readers may not know where Gothenburg is and how it looks like. Simple descriptions may be helpful. In addition, all the contours do not show any geographical and administrative boundaries. They may be also helpful to recognize where land, ocean, and ports are.

**Response to point 5:** We thank the reviewer for pointing out the need to further describe research domain in the part II manuscript. Therefore, we decided to add a figure of the research domain in the part II manuscript, and add some general information on the Gothenburg urban area. Due to the density of information, which is already provided in all contour plots in the manuscript, we decided not to add geographical references or administrative boundaries. Nevertheless, the additional figure on the research domain shall give guidance to recognize underlying geographical characteristics.

The following was added to the manuscript in a new section 2.1 on Gothenburg:

[Figure]

**Figure 1: The Gothenburg research domain. The light red grid indicates the domain extent and the horizontal grid-cell size of 250m. Red areas indicate port areas and grey lines indicated the city boundaries as given by the Copernicus Urban Atlas 2012 dataset. Maps are created with ArcGIS with underlying basemap sources Esri, HERE, Garmin, GEBCO, National Geographic, NOAA, and GIS User Community.**

"The city of Gothenburg is located on the western coast of Sweden, with about 0.57 million inhabitants and an area of 450 km2. The dominant wind direction in Gothenburg is south-west with average wind speed of 3.5 m s-1, indicating the major transport path from sea to the land, especially in summer. The geomorphology of the Gothenburg area is described as a fissure valley landscape dominated by a few large valleys in north-south and east-west directions. The major air pollution sources in Gothenburg are above all road traffic and industry, wood burning, shipping, agriculture, working machines and long-range transport (LRT) from the European continent and other parts of Sweden. The harbour and shipping activities are important emission sources and directly influences the urban air quality. The centre of the city is situated on the southern shore of the river Göta älv. The Port of Gothenburg receives between 6,000 and 6,500 calls per year and additional 600–700 ships pass to and from ports upstream and on the Göta älv. The port annually handles approximately 900,000 containers, 20 million tonnes of petroleum, and half a million Roll-on/roll-off (RoRo) units (Winnes et al., 2015). Passenger traffic in Gothenburg is also

very busy with 1.5 million passengers who ferry to and from Gothenburg to Denmark, Germany etc. on Stena Line ferries each year. This makes the port the largest cargo port in Scandinavia."

**Point 6**: Specific comments.

5  **Response to point 6:** Specific comments are answered hereunder.

P1, L27-30 Two expressions, "wide use of shore-site electricity" and "implementation of on-shore electricity", are confusing. They correspond to the same thing, right?

Yes. We aligned the expressions in the manuscript.

10  P3, L4-5 Critical not only for NO2 and O3 but also PM2.5 as mentioned above.

Yes. We added this in the manuscript.

P3, L10 Where is "this region"? It is not explicitly mentioned in preceding sentences.

The Baltic Sea region. We added this in the manuscript.

P5, Figure 1 This figure is not mentioned in any sentence in the manuscript.

We added a reference to figure 1 in the beginning of chapter 2.1.

P5, L4 What is the reason to couple TAPM and CMAQ instead of using TAPM or CMAQ only? Is that described in the Part

20  I paper?

There are some reasons to couple TAPM and CMAQ, which have been described in part I:

- TAPM allows for urban-scale pollutant concentration simulation due to the possibility and proven performance for simulation with resolutions below 1 km. Additionally TAPM takes into account detailed urban land-surface parametrizations.
25  - CMAQ was used to simulate regional-scale pollutant concentrations, while TAPM was used to simulate urban-scale pollutant concentrations. The coupling of CMAQ and TAPM allows for the consideration of consistent background concentrations, especially in future scenarios. In future scenarios, the simulations in CMAQ and TAPM rely on emission inventories, which both are based on the same assumptions for future trends.
- The regional CMAQ simulations by Karl et al. 2019 have been validated and compared to other regional CTM
30  simulations, showing good performance as part of the SHEBA project.

P5, L16 I think that evaluations cannot be done for the future year 2040.

Yes. We changed this in the manuscript.

35  P6, L9-14 Is it appropriate to use different horizontal resolutions for air pollution and meteorological fields? How to interpolate or extrapolate either of them?

The CTM TAPM consists of a meteorological module and a chemistry transport module. The chemistry transport module domain is nested in the meteorological module domain. When the horizontal grid resolution of the meteorological domain is 500 m, while the horizontal grid resolution of the CTM domain is 250 m, each of the four grid cells in the CTM domain applies the same meteorological values of the meteorological domain. Thus, there is no interpolation of meteorological values. Nevertheless, the CTM module treats each of the "finer" grid cells with 250 m independently in terms of transport and chemical reactions based on the "coarse" meteorological input in combination with the "neighbor" grid cells in the CTM domain.

We chose to apply a 500m resolution in the meteorological simulations after performing several tests to evaluate simulated meteorological against measurements (wind speed, wind direction, irradiation, and temperature). Meteorological simulations with a resolution of 250 m showed less statistical agreement with measurements than simulations with a resolution of 500 m. Thus, we decided to apply a meteorological horizontal grid resolution of 500 m.

P7, L22-23 I suppose that a scaling factor for combustion in industry for energy purposes is large because their VOC emissions are very low. Please check their emission amounts in GAINS. But that is not in the case in the emissions used in this study shown in Figure 2. It might be due to inconsistent definitions of sectors. Is there any appropriate reason explaining why their emissions significantly increase in Gothenburg?

Regarding VOC emissions, the situation in Gothenburg is largely influenced by two refineries, which are completely dominating the VOC emissions in the city. We have carefully evaluated the current contribution and future trend for these two sources and according to the trends in ECLIPSE/GAINS these sources are expected to increase in future. These trends were developed before the much more ambitious targets on reductions of $CO_2$ emissions were adopted, which would probably lead to different trends if developed today. This illustrated the large uncertainties in the scenario work.

P12, L5-6 I cannot understand differences between BAU2040 and EEDI2040 for fuel efficiencies. What kind of policies are assumed in each scenario? Why fuel efficiency is higher in BAU2040 than EEDI2040? What is the motivation to compare these two scenarios? Please add more explanations to clarify significance of EEDI2040 scenario.

We added additional information on the scenarios under Point 3 and Point 4, as well as in the manuscript, to clarify the differences.

P13, 3.3 Scenario setup. Were simulations conducted for twelve months in 2012?

Yes, we conducted simulations for twelve months in 2012 as described in Part I. We added this information in the manuscript.

P15, L4-5 Is 7% reduction a high potential? In fact, I cannot understand which part of Figure 4 this sentence describes.

This was a typo. It is meant to be 70% and is now corrected in the manuscript.

P15, L18-20 I cannot believe such high PM2.5 concentrations according to Figure 4. In addition, I think the units of absolute and relative differences of PM2.5 in Figure 4 are opposite.

These extremely high pollutant concentrations are the result of single point sources (mostly refineries), which have very high emissions and therefore lead to such high hourly maxima of $PM_{2.5}$ concentrations.

The units in figure 4 were flipped and are now corrected in the manuscript.

P17, L4 It is confusing to represent changes in negative contributions to O3 as "increasing".

We agree that this might be confusing. Nevertheless, it is consistent and in-line with the presentation of the results of other pollutants.

P20, L16-19 I think longer lifetime of secondary components in the atmosphere should be also one of reasons.

We agree. This can be considered as another reason and was added to the manuscript.

References Some references have no years. Particularly, it is difficult to distinguish IMO reports.

The references were again checked and corrected for missing years.

15  **Response to point 1:** We thank the Reviewer for providing a detailed evaluation of our study, the manuscript and the helpful comments and suggestions regarding the methodology used in our study.

**Point 2:** Please add general descriptions of Gothenburg, including its graphical locations. Moreover, please add longitudes, latitudes and geographical information for all the spatial distribution maps in the manuscript.

20  We thank the reviewer for pointing out the need to further describe research domain in the part II manuscript. Therefore, we decided to add a figure of the research domain in the part II manuscript, and add some general information on the Gothenburg urban area. Due to the density of information, which is already provided in all contour plots in the manuscript, we decided not to add geographical references or administrative boundaries. Nevertheless, the additional figure on the research domain shall give guidance to recognize underlying geographical characteristics.

25  The following was added to the manuscript in a new section 2.1 on Gothenburg:

[Figure]

**Figure 2: The Gothenburg research domain. The light red grid indicates the domain extent and the horizontal grid-cell size of 250m. Red areas indicate port areas and grey lines indicated the city boundaries as given by the Copernicus Urban Atlas 2012 dataset. Maps are created with ArcGIS with underlying basemap sources Esri, HERE, Garmin, GEBCO, National Geographic, NOAA, and GIS User Community.**

"The city of Gothenburg is located on the western coast of Sweden, with about 0.57 million inhabitants and an area of 450 km2. The dominant wind direction in Gothenburg is south-west with average wind speed of 3.5 m s-1, indicating the major transport path from sea to the land, especially in summer. The geomorphology of the Gothenburg area is described as a fissure valley landscape dominated by a few large valleys in north-south and east-west directions. The major air pollution sources in Gothenburg are above all road traffic and industry, wood burning, shipping, agriculture, working machines and long-range transport (LRT) from the European continent and other parts of Sweden. The harbour and shipping activities are important emission sources and directly influences the urban air quality. The centre of the city is situated on the southern shore of the river Göta älv. The Port of Gothenburg receives between 6,000 and 6,500 calls per year and additional 600–700 ships pass to and from ports upstream and on the Göta älv. The port annually handles approximately 900,000 containers, 20 million tonnes of petroleum, and half a million Roll-on/roll-off (RoRo) units (Winnes et al., 2015). Passenger traffic in Gothenburg is also

very busy with 1.5 million passengers who ferry to and from Gothenburg to Denmark, Germany etc. on Stena Line ferries each year. This makes the port the largest cargo port in Scandinavia. Annual analyses of air quality monitoring data Environmental Administration of City of Gothenburg show exceedances of both the target and the limit values for NO2 at several stations in Gothenburg in 2012 with decreasing trends towards exceedances of only the limit value at traffic stations in 2019. For PM10 the levels were well below the limit value but exceeding the target value in 2012 without any significant trend towards presence with exception of the urban background where slightly decreasing trend have been observed and the annual mean was bellow the target value of 15 µg/m³ the last 4 years. The measured concentration levels of PM2.5 have been bellow the target value without any significant trend at Gothenburg monitoring stations. Concentrations of ozone have a slightly increasing trend from year 2012 onwards and tend to exceed the limit values for maximum hourly and 8-h means at a number of occasions each year (Miljöförvaltningen, 2019 )."

**Point 2:** This study adopted meteorological field of 2012 in the simulation. The diffusion conditions may influence the impacts of emission reduction on air quality. So please add descriptions of the meteorological fields of 2012 to describe whether it is a year with good diffusion conditions or not. I suggest selecting a year of which the meteorological conditions are close to the climatological conditions, and then conduct the simulation.

**Response to point 2:** We thank the Reviewer for pointing out the need to clarify our decisions for the meteorological base year. This study has been conducted within the BONUS SHEBA project (Shipping and Environment of the Baltic Sea Region) where the impact of current and scenario emissions from ships on air quality have been investigated as a part of a holistic assessment framework for impacts of shipping on marine and coastal environment. The shipping-related air pollution has been investigated on a range of spatial scales with several chemistry-transport models: coarse spatial scale resolution was used for simulations in the European domain, finer resolution was used for the Baltic Sea (Karl et al., 2019b; Karl et al., 2019a), and city-scale simulations using high spatial resolution were used for several harbour cities (Ramacher et al., 2019). The present study (Part I) evaluates the contributions of regional and local shipping to the concentrations of $SO_2$, $NO_2$, $PM_{2.5}$, $O_3$ and secondary PM, as well as the human exposure and the associated health impacts in Gothenburg for year 2012.

All studies conducted are based on the reference year 2012. Based on the temperature anomalies and precipitation anomalies for the decade 2004–2014 for Baltic Proper, the year 2012 was chosen as the meteorological reference year for the CTM simulations in Part I of the Gothenburg study as well as in regional studies for current (2012) and future (2040) conditions and shipping scenarios. Year 2012 anomalies for 2 m temperature (±2 ∘C) and total precipitation (±25 mm) were closely aligned with the decadal average of the 2004–2014 period. The meteorological year 2012 was also used in CTM calculations of the future air quality situation to avoid complication of the interpretation of changes between the present-day and the future.

We added the information on 2012 representing a reference year for the region to the manuscript:

"Based on the temperature anomalies and precipitation anomalies for the decade 2004–2014 for Baltic Proper, the year 2012 was chosen as the meteorological reference year for the CTM simulations in Part I of the Gothenburg study as well as in regional studies for current (2012) and future (2040) conditions and shipping scenarios (Karl et al. 2019, Tang et al. 2020)."

**Point 3:** In Section 5, this study assessed the impact of future shipping on human health, including premature deaths because the decrease of ambient PM2.5, O3, and NO2. Exposures to PM2.5, NO2, and O3 can all lead to premature deaths due to respiratory diseases. So in Table3, I am wondering whether there are overlaps between the number of premature deaths due to PM2.5 with those due to NO2 and O3.

**Response to point 3:** We thank the reviewer for this comment. The health impact are presented for each pollutant separately and these impacts are not additive, In methodology part of Part 1 of this study (Tang et al., 2020) we state:

"The health impacts of some pollutants are correlated and that is why the premature deaths attributed to each pollutant cannot simply be added up. In particular, it has been estimated that adding premature deaths attributed to $PM_{2.5}$ to those attributed to $NO_2$ could result in double counting of around 30 % (WHO 2013a)."

**Point 4:** Minor comments.

**Response to point 4:** Minor comments are answered hereunder.

1. P6 Line10-15: Add more information for the simulation, including a figure to present the domains of the simulation, the period of the simulation, model spin-up, etc.

We added a new figure to the manuscript as introduced in our response to point 1. Additionally we added information on the model setup in the supplement, due to this information mostly given in the accompanying part 1 publication:

"

Table S4-1: City-scale model setup.

| | Domain | Spatial resolutions | Model / Database |
|---|---|---|---|
| Meteorology 2012 | 30 km × 30 km | 500 m | ECMWF ERA5 0.3˚ × 0.3˚, 21 layers |
| Background concentrations | 160 km × 96 km | 4 km × 4 km | CMAQ |
| Local shipping emissions 2012 | 30 km × 30 km | 250 m × 250 m | STEAM2 |
| Local traffic emissions 2012 | 30 km × 30 km | meters (line sources) | Miljöförvaltningen and HBEFA v. 3.2 |
| Local industrial, machines, wood burning and aviation etc. 2012 | 30 km × 30 km | 1 km × 1 km | SMED |

The period of the simulation is the year 2012 (introduced in the manuscript) and due to the rather fast chemistry on urban-scales, there is no model spin-up necessary. Tests with and without a model spin-up time of one week have shown differences in results below 0.1% for the first hours of simulated concentrations in the simulation period."

2. Cite Figure 1 in the manuscript, or delete it.

Figure 1 is now cited in the manuscript in 2.3.

3. Please show the spatial distribution of the emission inventories of 2012.

The spatial distribution of local shipping emissions has been shown in Paper I, which will be included in the supplement.

[Figure]

**Figure 2.** Annual local shipping emissions of **(a)** NO$_x$ and **(b)** PM$_{10}$ (equal to PM$_{2.5}$) from small vessels with a stack height below 36 m (assumed 15 m) and **(c)** NO$_x$ and **(d)** PM$_{10}$ from large vessels with high stack height above 36 m (assumed 36 m) in the Gothenburg area. Base map credits: © OpenStreetMap contributors 2020. Distributed under a Creative Commons BY-SA License.

Moreover, the following figure shows the spatial distribution of local emissions from road traffic and industrial point sources. In addition, other emissions such as domestic heating, working and off-road machinery etc. expressed as grid sources in the model. The map on spatial distribution of emissions is now included in the supplement.

[Figure]

The spatial distribution of local emissions from road traffic (red lines), industrial point sources (green circles), and other sources (yellow lines).

4. In Figure 4, the unit for figure in row3 column 3 should be ug/m3; the unit for figure in row3 column 4 should be %.

We changed the figure accordingly. Thank you for your detailed examination of our manuscript.

**Figure 3: The Gothenburg research domain. The light red grid indicates the domain extent and the horizontal grid-cell size of 250m. Red areas indicate port areas and grey lines indicated the city boundaries as given by the Copernicus Urban Atlas 2012 dataset. Maps are created with ArcGIS with underlying basemap sources Esri, HERE, Garmin, GEBCO, National Geographic, NOAA, and GIS User Community.**

**2.1 2 Global- to urban-scale CTM system setup**

[revised manuscript text omitted]

**3.2.2 Future scenario EEDI2040**

20   As the scenario work revealed that energy effectivization has large impact on emissions in the target year, encompassing at the same time large uncertainty, we have chosen to include an alternative scenario with different effectivization level. 
[revised manuscript text omitted]

In 2040 simulations the air quality situation in the city improves considerably and the concentrations will be below the air

20  quality limit and target values both for $NO_2$ and $PM_{2.5}$ in the city, even if the potential underestimates of the model system is accounted for (Tang et al., 2020). From the static point of view of year 2040 conclusion that additional measures to reduce air pollution levels beyond those in the BAU scenario would not be necessary could be drawn. However, in perspective taking also the temporal development into consideration, the measures reducing concentrations of $NO_2$ will be implemented only in a slow pace and full impact of many of them, especially those targeting shipping, will first be seen in the time horizon of 2040.

25  The local measures could be, on the other hand, implemented faster and the significant reduction of $NO_2$ concentrations from implementation of on-shore electricity could thus reduce the time before the air quality targets are met in the city.

[revised manuscript text omitted]

15   Miljöförvaltningen: Luften i Göteborg: Årsrapport 2019: https://goteborg.se/wps/wcm/connect/10808596-7471-4e9e-af8a-2f7f517140af/R+2020_12+Luften+i+G%C3%B6teborg+-+%C3%A5rsrapport+2019.pdf?MOD=AJPERES, last access: 29 June 2020.

Ramacher, M. O. P. and Karl, M.: Integrating Modes of Transport in a Dynamic Modelling Approach to Evaluate Population Exposure to Ambient NO2 and PM2.5 Pollution in Urban Areas, IJERPH, 17, 2099, doi:10.3390/ijerph17062099, 2020.

20   Ramacher, M. O. P., Karl, M., Aulinger, A., and Bieser, J.: Population Exposure to Emissions from Industry, Traffic, Shipping and Residential Heating in the Urban Area of Hamburg, in: Air Pollution Modeling and its Application XXVI, Mensink, C., Gong, W., and Hakami, A. (Eds.), Springer Proceedings in Complexity, Springer International Publishing, Cham, 177–183, 2020.

Ramacher, M. O. P., Karl, M., Bieser, J., Jalkanen, J.-P., and Johansson, L.: Urban population exposure to $NO_x$ emissions
25   from local shipping in three Baltic Sea harbour cities – a generic approach, Atmos. Chem. Phys. Discuss., 1–45, doi:10.5194/acp-2019-127, 2019a.

Ramacher, M. O. P., Karl, M., Bieser, J., Jalkanen, J.-P., and Johansson, L.: Urban population exposure to NOx emissions from local shipping in three Baltic Sea harbour cities – a generic approach, Atmos. Chem. Phys., 19, 9153–9179, doi:10.5194/acp-19-9153-2019, 2019b.

30   Reis, S., Liška, T., Vieno, M., Carnell, E. J., Beck, R., Clemens, T., Dragosits, U., Tomlinson, S. J., Leaver, D., and Heal, M. R.: The influence of residential and workday population mobility on exposure to air pollution in the UK, Environment international, 121, 803–813, doi:10.1016/j.envint.2018.10.005, 2018.

Rockel, B., Will, A., and Hense, A.: The Regional Climate Model COSMO-CLM (CCLM), metz, 17, 347–348, doi:10.1127/0941-2948/2008/0309, 2008.

Sillman, S.: The relation between ozone, NOx and hydrocarbons in urban and polluted rural environments, Atmospheric Environment, 33, 1821–1845, doi:10.1016/S1352-2310(98)00345-8, 1999.

Smith, J. D., Mitsakou, C., Kitwiroon, N., Barratt, B. M., Walton, H. A., Taylor, J. G., Anderson, H. R., Kelly, F. J., and Beevers, S. D.: London Hybrid Exposure Model: Improving Human Exposure Estimates to NO2 and PM2.5 in an Urban Setting, Environmental science & technology, 50, 11760–11768, doi:10.1021/acs.est.6b01817, 2016.

Soares, J., Kousa, A., Kukkonen, J., Matilainen, L., Kangas, L., Kauhaniemi, M., Riikonen, K., Jalkanen, J.-P., Rasila, T., Hänninen, O., Koskentalo, T., Aarnio, M., Hendriks, C., and Karppinen, A.: Refinement of a model for evaluating the population exposure in an urban area, Geosci. Model Dev., 7, 1855–1872, doi:10.5194/gmd-7-1855-2014, 2014.

Sofiev, M., Winebrake, J. J., Johansson, L., Carr, E. W., Prank, M., Soares, J., Vira, J., Kouznetsov, R., Jalkanen, J.-P., and Corbett, J. J.: Cleaner fuels for ships provide public health benefits with climate tradeoffs, Nature communications, 9, 406, doi:10.1038/s41467-017-02774-9, 2018.

Tang, L., Ramacher, M. O. P., Moldanova, J., Matthias, V., Karl, M., and Johansson, L.: The impact of ship emissions on air quality and human health in the Gothenburg area – Part 1: Current situtation, Atmos. Chem. Phys. Discuss., in preparation, 2020.

Transport administration: Prognos för persontrafiken 2040: Trafikverkets Basprognoser 2016, Report 2016:062, Transport administration (Trafikverket), 2016.

Transport administration: Prognos för persontrafiken 2040: Trafikverkets Basprognoser 2018-04-01, Report 2018:089, Transport administration (Trafikverket), 2018.

Zandersen, M., Hyytiäinen, K., Meier, H. E. M., Tomczak, M. T., Bauer, B., Haapasaari, P. E., Olesen, J. E., Gustafsson, B. G., Refsgaard, J. C., Fridell, E., Pihlainen, S., Le Tissier, M. D. A., Kosenius, A.-K., and van Vuuren, D. P.: Shared socio-economic pathways extended for the Baltic Sea: exploring long-term environmental problems, Reg Environ Change, 19, 1073–1086, doi:10.1007/s10113-018-1453-0, 2019.